# Downregulation of semaphorin 4A in keratinocytes reflects the features of non-lesional psoriasis

Miki Kume[1], Hanako Koguchi-Yoshioka[1,2], Shuichi Nakai[1,3], Yutaka Matsumura[1], Atsushi Tanemura[1], Kazunori Yokoi[1], Shoichi Matsuda[1,3], Yuumi Nakamura[1,4], Naoya Otani[5], Mifue Taminato[5], Koichi Tomita[5,6], Tateki Kubo[5], Mari Wataya-Kaneda[1,2], Atsushi Kumanogoh[7], Manabu Fujimoto[1], Rei Watanabe[1,8]*

[1]Department of Dermatology, Course of Integrated Medicine, Graduate School of Medicine, Osaka University, Osaka, Japan; [2]Department of Neurocutaneous Medicine, Division of Health Sciences, Graduate School of Medicine, Osaka University, Osaka, Japan; [3]Research Department, Maruho Co, Ltd., Kyoto, Japan; [4]Cutaneous Allergy and Host Defense, Immunology Frontier Research Center (iFReC), Osaka University, Osaka, Japan; [5]Department of Plastic Surgery, Course of Organ Regulation Medicine, Graduate School of Medicine, Osaka University, Osaka, Japan; [6]Department of Plastic and Reconstructive Surgery, Kindai University, Osaka, Japan; [7]Department of Respiratory Medicine and Clinical Immunology, Course of Internal Medicine, Graduate School of Medicine, Osaka University, Osaka, Japan; [8]Department of Medicine for Cutaneous Immunological Diseases, Course of Integrated Medicine, Graduate School of Medicine, Osaka University, Osaka, Japan

*For correspondence: rwatanabe@derma.med.osaka-u.ac.jp

## eLife Assessment

This paper advances an **important** new concept in psoriasis pathogenesis and implicates Sema4a as a homeostatic regulator that is highly epithelial-specific. The findings are **convincing** and lend support for the biology described here as a mechanism with therapeutic implications.

**Abstract** Psoriasis is a multifactorial disorder mediated by IL-17-producing T cells, involving immune cells and skin-constituting cells. Semaphorin 4A (Sema4A), an immune semaphorin, is known to take part in T helper type 1/17 differentiation and activation. However, Sema4A is also crucial for maintaining peripheral tissue homeostasis and its involvement in skin remains unknown. Here, we revealed that while Sema4A expression was pronounced in psoriatic blood lymphocytes and monocytes, it was downregulated in the keratinocytes of both psoriatic lesions and non-lesions compared to controls. Imiquimod application induced more severe dermatitis in Sema4A knockout (KO) mice compared to wild-type (WT) mice. The naïve skin of Sema4A KO mice showed increased T cell infiltration and IL-17A expression along with thicker epidermis and distinct cytokeratin expression compared to WT mice, which are hallmarks of psoriatic non-lesions. Analysis of bone marrow chimeric mice suggested that Sema4A expression in keratinocytes plays a regulatory role in imiquimod-induced dermatitis. The epidermis of psoriatic non-lesion and Sema4A KO mice demonstrated mTOR complex 1 upregulation, and the application of mTOR inhibitors reversed the skewed expression of cytokeratins in Sema4A KO mice. Conclusively, Sema4A-mediated signaling cascades can be triggers for psoriasis and targets in the treatment and prevention of psoriasis.

## Introduction

While the infiltration of immune cells into skin plays a critical role in the development of psoriasis, as evidenced by interleukin (IL)-23/IL-17 axis (*Fitch et al., 2007*; *Hawkes et al., 2018*; *Kim and Krueger, 2017*), recent studies have revealed that cells constructing skin structure, such as keratinocytes, fibroblasts, and endothelial cells, also play pivotal roles in the development (*Heidenreich et al., 2009*; *Lowes et al., 2013*; *Zhang et al., 2023*) and maintenance (*Arasa et al., 2019*; *Francis et al., 2024*; *Li et al., 2023*; *Ma et al., 2023*; *Tan et al., 2015*; *Zhu et al., 2020*) of psoriasis. Among these cells, keratinocytes function as a barrier and produce cytokines, chemokines, and antimicrobial peptides against foreign stimuli, resulting in the activation of immune cells (*Ni and Lai, 2020*; *Zhou et al., 2022*).

Semaphorins were initially identified as guidance cues in neural development (*Kolodkin et al., 1993*; *Pasterkamp and Kolodkin, 2003*) but are now regarded to play crucial roles in other physiological processes including angiogenesis (*Iragavarapu-Charyulu et al., 2020*; *Serini et al., 2009*), tumor microenvironment (*Hui et al., 2019*; *Jiang et al., 2022*; *Nakayama et al., 2018*; *Rajabinejad et al., 2020*), and immune systems (*Garcia, 2019*; *Kanth et al., 2021*; *Naito and Kumanogoh, 2023b*). Semaphorin 4A (Sema4A), one of the immune semaphorins, plays both pathogenic and therapeutic roles in autoimmune diseases (*Cavalcanti et al., 2020*; *He et al., 2023*), allergic diseases (*Maeda et al., 2019*), and cancer (*Iyer and Chapoval, 2018*; *Liu et al., 2018*; *Naito et al., 2023a*; *Pan et al., 2016*). While Sema4A expression on T cells is essential for T helper type 1 differentiation in the murine *Propionibacterium acnes*-induced inflammation model and delayed-type hypersensitivity model (*Kumanogoh et al., 2005*), Sema4A amplifies only T helper type 17 (Th17)-mediated inflammation in the effector phase of murine experimental autoimmune encephalomyelitis (*Koda et al., 2020*). Accordingly, in multiple sclerosis, high serum Sema4A levels correlate with the elevated serum IL-17A, earlier disease onset, and increased disease severity (*Koda et al., 2020*; *Nakatsuji et al., 2012*). In anti-tumor immunity, research involving human samples and murine models suggests that Sema4A expressed in cancer cells and regulatory T cells promotes tumor progression (*Delgoffe et al., 2013*; *Liu et al., 2018*; *Pan et al., 2016*), while other reports reveal that Sema4A in cancer cells and dendritic cells bolsters anti-tumor immunity by enhancing CD8 T cell activity (*Naito et al., 2023a*; *Suga et al., 2021*). In addition to these roles in immune reactions, mice with a point mutation in Sema4A develop retinal degeneration, suggesting that Sema4A is also crucial for peripheral tissue homeostasis (*Nojima et al., 2013*). These reports suggest that the role of Sema4A can differ based on the disease, phase, and involved cells.

Herein, we investigated the roles of Sema4A in the pathogenesis of psoriasis by analyzing skin and blood specimens from psoriatic subjects and using a murine model.

## Results

### Epidermal Sema4A expression is downregulated in psoriasis

The analysis of previously published single-cell RNA-sequencing data from control (Ctl) and psoriatic lesional (L) skin specimens (*Kim et al., 2023*) revealed detectable expression of *SEMA4A* in keratinocytes, dendritic cells, and macrophages in both Ctl and L. *SEMA4A* expression was low in neural crest-like cells, fibroblasts, CD4 T cells, CD8 T cells, NK cells, and plasma cells, making the comparison of expression levels impractical (*Figure 1A and B*; *Figure 1—figure supplement 1A–C*). Dendritic cells and macrophages showed comparable *SEMA4A* expression levels between Ctl and L (*Figure 1C*). The adjusted p-value (padj) for *SEMA4A* in keratinocytes between Ctl and L was $2.83\times10^{-39}$, indicating a statistically significant difference despite not being visually prominent in the volcano plot, which shows comprehensive differential gene expression in keratinocytes (*Figure 1C*; *Figure 1—figure supplement 1D*).

Immunohistochemistry of Ctl and psoriasis demonstrated Sema4A expression in keratinocytes (*Figure 1D*). The staining intensity of Sema4A in epidermis was significantly lower in both non-lesions (NL) and L than in Ctl (*Figure 1E*). Relative mRNA expression of *SEMA4A* was also decreased in the epidermis of NL and L compared to Ctl, while it remained comparable in the dermis (*Figure 1F*). In contrast, the proportions of Sema4A-positive cells were significantly higher in blood CD4 and CD8 T cells, and monocytes in psoriasis compared to Ctl (*Figure 1G*; *Figure 1—figure supplement 2*). Serum Sema4A levels, measured by enzyme-linked immunosorbent assay (ELISA), were comparable

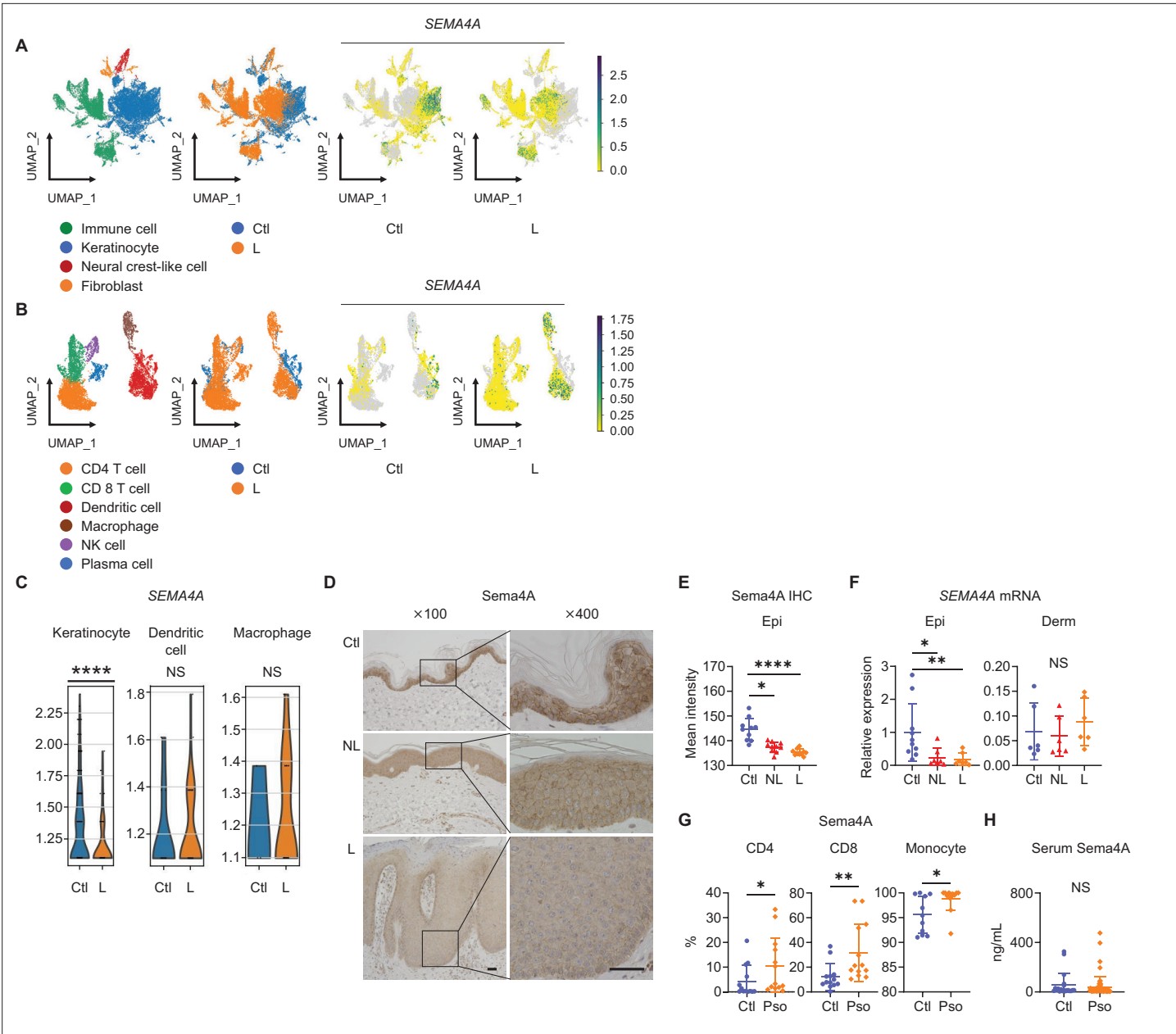

**Figure 1.** Epidermal Sema4A expression is downregulated in psoriasis. (**A**) UMAP plots, generated from single-cell RNA-sequencing data (GSE220116), illustrate cell distributions from control (Ctl) and psoriatic lesion (L) samples (n=10 for Ctl, n=11 for L). (**B**) Subclustering of immune cells. (**C**) *SEMA4A* expression in keratinocytes, dendritic cells, and macrophages. ****padj<0.001. NS, not significant. Analyzed using Python and cellxgene VIP. (**D**) Representative immunohistochemistry and magnified views showing Sema4A expression in Ctl, psoriatic non-lesion (NL), and L. Scale bar = 50 µm. (**E**) Mean epidermal (Epi) Sema4A intensity in immunohistochemistry (n=10 per group). Each dot represents the average intensity from 5 unit areas per sample. (**F**) Relative *SEMA4A* expression in Epi (n=10 for Ctl, n=7 for L and NL) and dermis (Derm, n=6 per group). (**G**) Proportions of Sema4A-expressing cells in blood CD4 T cells (left), CD8 T cells (middle), and monocytes (right) from Ctl and psoriatic (Pso) patients (n=13 per group in CD4 and CD8, n=11 for Ctl and n=13 for Pso in monocytes). (**H**) Serum Sema4A levels in Ctl (n=20) and Pso (n=60). (**E–H**) *p<0.05, **p<0.01, ****p<0.0001. NS, not significant. The error bars represent the standard deviation.

The online version of this article includes the following source data and figure supplement(s) for figure 1:

**Source data 1.** Excel file containing quantitative data for *Figure 1*.

**Figure supplement 1.** Sema4A is downregulated in the keratinocytes of lesional psoriasis in the single-cell RNA-sequencing data.

**Figure supplement 1—source data 1.** Excel file containing quantitative data for *Figure 1—figure supplement 1*.

**Figure supplement 2.** Gating strategy in flow cytometry.

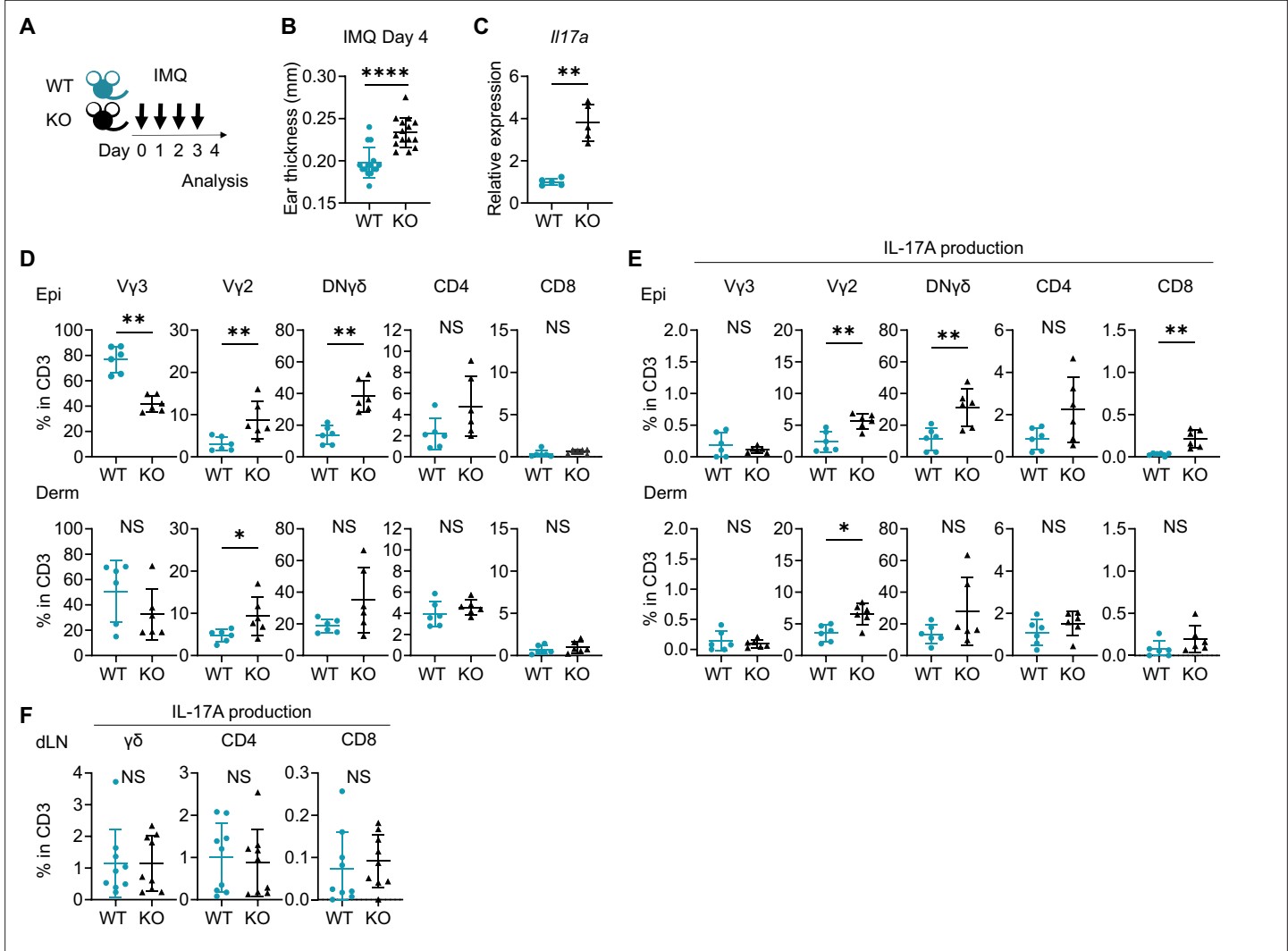

**Figure 2.** Imiquimod (IMQ)-induced psoriasis-like dermatitis is augmented in Sema4A knockout (KO) mice. (**A**) Experimental scheme. Wild-type (WT, green) mice and Sema4A KO (black) mice were treated with 10 mg/ear of 5% IMQ for 4 consecutive days. Samples for flow cytometry analysis were collected on day 4. (**B**) Ear thickness of WT mice and KO mice on day 4 (n=15 per group). (**C**) Relative expression of *Il17a* in epidermis (n=5 per group). (**D, E**) The percentages of Vγ3, Vγ2, Vγ2-Vγ3-γδ (DNγδ), CD4, and CD8 T cells (**D**) and those with IL-17A production (**E**) in CD3 fraction in the Epi (top) and Derm (bottom) of WT and KO ears (n=6 per group, each dot represents the average of 4 ear specimens). (**F**) The percentages of IL-17A-producing γδ, CD4, and CD8 T cells in CD3 fraction in skin-draining lymph nodes (dLN) (n=9 per group). (**B–F**) *p<0.05, **p<0.01, ****p<0.0001. NS, not significant. The error bars represent the standard deviation.

The online version of this article includes the following source data and figure supplement(s) for figure 2:

**Source data 1.** Excel file containing quantitative data for *Figure 2*.

**Figure supplement 1.** Gating strategy in flow cytometry.

**Figure supplement 2.** IL-23-mediated psoriasis-like dermatitis is augmented in Sema4A knockout (KO) mice.

**Figure supplement 2—source data 1.** Excel file containing quantitative data for *Figure 2—figure supplement 2*.

between Ctl and psoriasis (*Figure 1H*). These findings demonstrate that the expression profile of Sema4A in psoriasis varies across cell types.

## Psoriasis-like dermatitis is augmented in Sema4A KO mice

When psoriasis-like dermatitis was induced in wild-type (WT) mice and Sema4A knockout (KO) mice by imiquimod (IMQ) application on ears (*Figure 2A*), ear swelling on day 4 was more pronounced in Sema4A KO mice (*Figure 2B*) with upregulated *Il17a* gene expression (*Figure 2C*). Flow cytometry analysis of cells isolated from the ears revealed increased proportions of Vγ2+ T cells,

Vγ2⁻Vγ3⁻ double-negative (DN) γδ T cells, and IL-17A-producing cells of those fractions in Sema4A KO epidermis (*Figure 2D and E*; *Figure 2—figure supplement 1*). In Sema4A KO dermis, there was also an increase in the proportions of Vγ2⁺ T cells and IL-17A-producing Vγ2⁺ T cells (*Figure 2D and E*). These results suggest that Sema4A deficiency in mice accelerates psoriatic profile. IL-17A-producing T cells in skin-draining lymph nodes (dLN) remained comparable between WT mice and Sema4A KO mice (*Figure 2F*). Though the IMQ model is well established and valuable murine psoriatic model (*van der Fits et al., 2009*), the vehicle of IMQ cream can activate skin inflammation that is independent of Toll-like receptor 7, such as inflammasome activation, keratinocyte death, and interleukin-1 production (*Walter et al., 2013*). This suggests that the IMQ model involves complex pathway. Therefore, we subsequently induced IL-23-mediated psoriasis-like dermatitis (*Figure 2—figure supplement 2A*), a much simpler murine psoriatic model, because IL-23 is thought to play a central role in psoriasis pathogenesis (*Krueger et al., 2007*; *Lee et al., 2004*). Although ear swelling on day 4 was comparable between WT mice and Sema4A KO mice (*Figure 2—figure supplement 2B*), the epidermis, but not the dermis, was significantly thicker in Sema4A KO mice compared to WT mice (*Figure 2—figure supplement 2C*). We found that the proportion of CD4 T cells among T cells was significantly higher in Sema4A KO mice compared to WT mice, while the proportion of Vγ2 and DNγδ T cells among T cells was comparable between them (*Figure 2—figure supplement 2D*). On the other hand, focusing on IL-17A-producing cells, the proportion of IL-17A-producing Vγ2 and DNγδ T cells in CD3 fraction in the epidermis was significantly higher in Sema4A KO mice, consistent with the results from IMQ-induced psoriasis-like dermatitis (*Figure 2—figure supplement 2E*).

## Sema4A in keratinocytes may play a role in preventing murine psoriasis-like dermatitis

To investigate the cells responsible for the augmented ear swelling in Sema4A KO mice, bone marrow chimeric mice were next analyzed (*Figure 3A*). Since it has already been reported that bone marrow cells contain keratinocyte stem cells (*Harris et al., 2004*; *Wu et al., 2010*), we confirmed that epidermis of mice deficient in non-hematopoietic Sema4A (WT→KO) showed no obvious detection of *Sema4a*, thereby ruling out the impact of donor-derived keratinocyte stem cells infiltrating the host epidermis (*Figure 3—figure supplement 1A*). WT→KO mice displayed more pronounced ear swelling than mice with intact Sema4A expression (WT→WT) following IMQ application (*Figure 3B*). Similarly, mice with a systemic deficiency of Sema4A (KO→KO) showed severe ear swelling compared to mice deficient in hematopoietic Sema4A (KO→WT) (*Figure 3B*). Ear swelling was comparable between WT→WT mice and KO→WT mice (*Figure 3B*). Flow cytometry analysis revealed increased infiltration of IL-17A-producing DNγδ T cells in the epidermis, as well as Vγ2⁺ T cells and IL-17A-producing Vγ2⁺ T cells in the dermis, in WT→KO mice compared to WT→WT mice (*Figure 3C*; *Figure 3—figure supplement 1B*). These findings suggest that non-hematopoietic cells, possibly keratinocytes, are primarily responsible for the increased IMQ-induced Sema4A KO mice ear swelling.

## Sema4A KO epidermis is thicker than WT epidermis with increased γδ T17 infiltration

Even without IMQ application, Sema4A KO ears turned out to be slightly but significantly thicker than WT ears on week 8 while their appearance remained normal (*Figure 4A*). While epidermal thickness of back skin was comparable at birth (*Figure 4B*), on week 8, epidermis of Sema4A KO back and ear skin was notably thicker than that of WT mice (*Figure 4B*), suggesting that acanthosis in Sema4A KO mice is accentuated post-birth. Dermal thickness remained comparable between WT mice and Sema4A KO mice at both times (*Figure 4B*). The epidermis of WT ear at week 8 showed significantly higher *Sema4a* mRNA expression compared to the dermis (*Figure 4C*). Based on these observations, Sema4A appears to play a more pronounced role in epidermis than in dermis.

Sema4A KO epidermis exhibited increased expression of *Ccl20*, *Tnfa*, and *Il17a* and a trend of upregulation of *S100a8* compared to WT epidermis (*Figure 4—figure supplement 1A*). These differences were not observed in dermis (*Figure 4—figure supplement 1A*). Flow cytometry analysis revealed increased infiltration of γδ T cells in Sema4A KO ear (*Figure 4—figure supplement 1B*). These cells predominantly expressed resident memory T cell (T_RM)-characteristic molecules, CD69 and CD103 (*Figure 4—figure supplement 1B*). Sema4A KO skin also had a higher number of T_RM in both CD4 and CD8 T cells (*Figure 4—figure supplement 1B*). The percentages of Vγ2⁺ T cells, DNγδ T

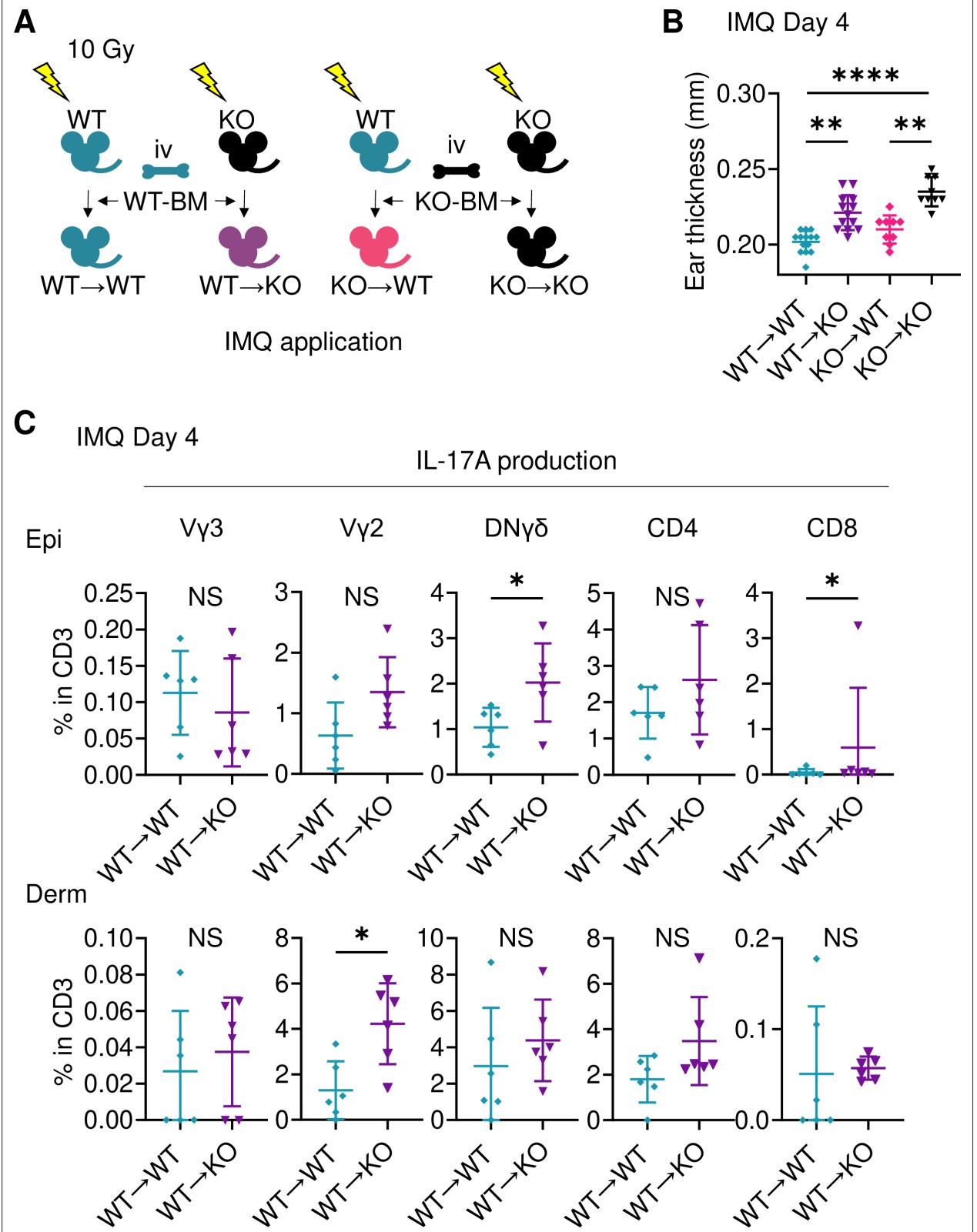

**Figure 3.** Sema4A in keratinocytes may play a role in preventing murine psoriasis-like dermatitis. (**A**) Experimental scheme for establishing BM chimeric mice. (**B**) Imiquimod (IMQ) day 4 ear thickness in the mice with the indicated genotypes (n=14 for WT→WT, n=13 for WT→KO, n=9 for KO→WT, n=9 for KO→KO). (**C**) The percentages of IL-17A-producing Vγ3, Vγ2, DNγδ, CD4, and CD8 T cells in CD3 fraction from IMQ day 4 Epi (top) and Derm (bottom)

*Figure 3 continued on next page*

*Figure 3 continued*

of the ears from WT→ WT mice and WT→ KO mice (n=6 per group). Each dot represents the average of 4 ear specimens. (**B, C**) *p<0.05, **p<0.01, ****p<0.0001. NS, not significant. The error bars represent the standard deviation.

The online version of this article includes the following source data and figure supplement(s) for figure 3:

**Source data 1.** Excel file containing quantitative data for *Figure 3*.

**Figure supplement 1.** T cells' fractions infiltrating in the chimeric mice ear.

**Figure supplement 1—source data 1.** Excel file containing quantitative data for *Figure 3—figure supplement 1*.

cells, and CD8 T cells in epidermis were higher in Sema4A KO mice than in WT mice, which was not the case in dermis (*Figure 4—figure supplement 2A*). The proportion of Vγ3+ dendritic epidermal T cells was comparable between WT mice and Sema4A KO mice (*Figure 4—figure supplement 2A*). Epidermal Vγ2+ T cells and DNγδ T cells in Sema4A KO mice showed higher IL-17A-producing capability (*Figure 4D*), while IFNγ and IL-4 production was comparable between WT mice and Sema4A KO mice (*Figure 4—figure supplement 2B and C*). Conversely, the frequency of IL-17A-producing T cells from dLN was comparable (*Figure 4E*). The production of IL-17A and IFNγ from splenic T cells under T17-polarizing conditions remained consistent between WT mice and Sema4A KO mice (*Figure 4—figure supplement 3*).

Taken together, it is suggested that T17 cells are specifically upregulated in epidermis, indicating that the epidermal microenvironment plays a pivotal role in facilitating the increased T cell infiltration observed in naïve Sema4A KO mice.

## Sema4A KO skin shares features with human psoriatic NL

Previous literatures have identified certain features common to psoriatic L and NL, such as thickened epidermis (*Figure 5—figure supplement 1*; *Gallais Sérézal et al., 2019*), CCL20 upregulation (*Gallais Sérézal et al., 2019*), and accumulation of IL-17A-producing T cells (*Vo et al., 2019*), which were detected in Sema4A KO mice.

Gene expression analysis using public RNA-sequencing data (*Tsoi et al., 2019*) with RaNAseq (*Prieto and Barrios, 2019*) showed upregulation of keratinization and antimicrobial peptide genes in NL compared to Ctl (*Figure 5A*). Gene Ontology analysis highlighted an upregulation in biological processes, predominantly in peptide cross-linking involved in epidermis formation and keratinocyte differentiation, with a secondary increase in the defense response to viruses in NL (*Figure 5B*). While the expression of *Keratin (KRT) 10* was comparable between NL and Ctl, upregulation in *KRT5*, *KRT14*, and *KRT16* was observed in NL (*Figure 5C*).

In the murine model, relative expression levels of *Krt10*, *Krt14*, *Krt16*, and *Filaggrin* were elevated in Sema4A KO epidermis (*Figure 5D*). Immunofluorescence analysis showed that Sema4A KO epidermis had a higher density of keratinocytes positive for Krt5, Krt10, Krt14, and Krt16 compared to WT epidermis (*Figure 5E*). This upregulation was not observed in back skin at birth (*Figure 5—figure supplement 2*). Comparable transepidermal water loss between WT mice and Sema4A KO mice indicated preserved skin barrier function in Sema4A KO mice (*Figure 5F*).

Based on these results, it is implied that the epidermis of human psoriatic NL and Sema4A KO mice exhibit shared pathways, potentially leading to an acanthotic state. Combined with the observed acanthosis and increased T17 infiltration in Sema4A KO mice, Sema4A KO skin is regarded to demonstrate the features characteristic of human psoriatic NL.

## mTOR signaling is upregulated in the epidermis of psoriatic NL and Sema4A KO mice

Previous reports have shown that mTOR pathway plays a critical role in maintaining epidermal homeostasis, as evidenced by mice with keratinocyte-specific deficiencies in Mtor, Raptor, or Rictor, which exhibit a hypoplastic epidermis with impaired differentiation and barrier formation (*Asrani et al., 2017*; *Ding et al., 2016*; *Ding et al., 2020*). We thus investigated mTOR pathway in both human and murine epidermis.

Immunohistochemical analyses highlighted the increase in phospho-S6 (p-S6), indicating the upregulation of mTOR complex (C) 1 signaling, in the epidermal upper layers of both L and NL compared to Ctl (*Figure 6A*). The activation of mTORC2 in the epidermis, inferred from phospho-Akt (p-Akt), was

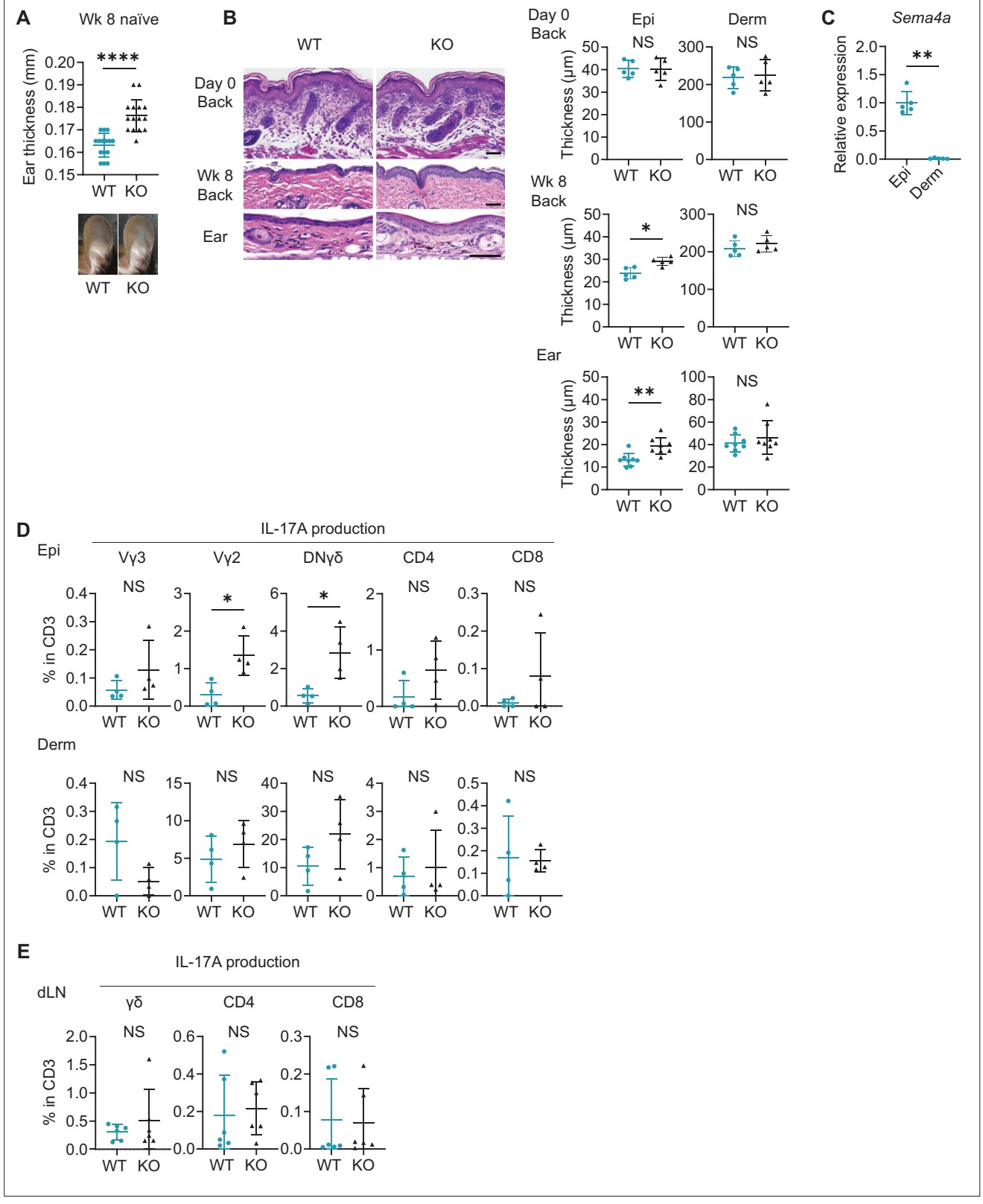

**Figure 4.** Naïve Sema4A knockout (KO) epidermis is thicker than wild-type (WT) epidermis with increased γδ T17 infiltration. (**A**) Ear thickness of WT mice and KO mice at week (Wk) 8 (n=15 per group) and representative images. (**B**) Left: representative hematoxylin and eosin staining of day 0 back and Wk 8 back and ear. Scale bar = 50 μm. Right: Epi and Derm thickness in day 0 back (n=5) and Wk 8 back (n=5) and ear (n=8). (**C**) Relative *Sema4a* expression in WT Epi and Derm (n=5 per group). (**D**) The percentages of the IL-17A-producing Vγ3, Vγ2, DNγδ, CD4, and CD8 T cells in CD3 fraction

*Figure 4 continued on next page*

*Figure 4 continued*

(n=4 per group) in Epi (top) and Derm (bottom). Each dot represents the average of 4 ear specimens. (**E**) The graphs showing the percentages of IL-17A-producing γδ, CD4, and CD8 T cells in CD3 fraction from draining LN (dLN) of WT mice and Sema4A KO mice (n=6 per group). (**A–E**) *p<0.05, **p<0.01, ****p<0.0001. NS, not significant. The error bars represent the standard deviation.

The online version of this article includes the following source data and figure supplement(s) for figure 4:

**Source data 1.** Excel file containing quantitative data for *Figure 4*.

**Figure supplement 1.** Naive Sema4A knockout (KO) skin shows upregulation of psoriasis-related genes and an increase in resident memory T cells.

**Figure supplement 1—source data 1.** Excel file containing quantitative data for *Figure 1—figure supplement 1*.

**Figure supplement 2.** Expression of IFNγ and IL-4 is comparable between naive wild-type (WT) and Sema4A knockout (KO) skin.

**Figure supplement 2—source data 1.** Excel file containing quantitative data for *Figure 4—figure supplement 2*.

**Figure supplement 3.** Comparable T17 differentiation potential under Th17-skewing conditions between wild-type (WT) mice and Sema4A knockout (KO) mice.

**Figure supplement 3—source data 1.** Excel file containing quantitative data for *Figure 4—figure supplement 3*.

scarcely detectable in L, NL, and Ctl (*Figure 6A*). In the epidermis of WT mice and Sema4A KO mice, the upregulation of both mTORC1 and mTORC2 signaling was observed in Sema4A KO epidermis by immunohistochemistry (*Figure 6B*). Western blot (WB) analysis showed upregulated mTORC1 signaling in Sema4A KO epidermis compared to WT epidermis (*Figure 6C*). The enhancement of mTORC1 and mTORC2 signaling became obvious in the Sema4A KO epidermis after developing psoriatic dermatitis by IMQ application (*Figure 6D*).

## Inhibition of mTOR signaling modulates cytokeratin expression in Sema4A KO mice

To investigate the contribution of mTORC1 and mTORC2 signaling in the development of psoriatic features in the Sema4A KO epidermis, mTORC1 inhibitor rapamycin and mTORC2 inhibitor JR-AB2-011 were intraperitoneally applied to Sema4A KO mice for 14 days. Although epidermal thickness remained unchanged by the inhibitors (*Figure 7A and B*), relative gene expression of *Krt5* was significantly upregulated and that of *Krt16* was significantly downregulated after rapamycin application (*Figure 7C*). While the upregulation of *Il17a* in Sema4A KO epidermis was not clearly modified by rapamycin (*Figure 7C*), immunofluorescence revealed the decrease in the number of CD3 T cells in Sema4A KO epidermis by rapamycin (*Figure 7D*). We additionally conducted topical application of rapamycin gel and vehicle gel on the left and right ears of Sema4A KO mice, respectively. Although there were no detectable changes in epidermal thickness and epidermal T cell counts, the upregulation of *Krt5* and downregulation of *Krt16* was observed again (*Figure 7—figure supplement 1*). Conversely, the application of JR-AB2-011 resulted in decreased expression of *Krt5*, *Krt10*, and *Krt14* with a trend toward increased *Krt16* expression (*Figure 7E*). JR-AB2-011 did not influence the number of infiltrating T cells in the epidermis (*Figure 7F*). Next, we investigated whether intraperitoneal rapamycin treatment effectively downregulates inflammation in the IMQ-induced murine model of psoriasis in Sema4A KO mice (*Figure 7—figure supplement 2A*). Rapamycin significantly reduced epidermal thickness compared to vehicle treatment (*Figure 7—figure supplement 2B*). Additionally, rapamycin treatment downregulated the expression of *Krt10*, *Krt14*, and *Krt16* (*Figure 7—figure supplement 2C*). While the upregulation of *Il17a* in the Sema4A KO epidermis in IMQ model was not clearly modified by rapamycin (*Figure 7—figure supplement 2C*), immunofluorescence revealed a decrease in the number of CD3 T cells in Sema4A KO epidermis by rapamycin (*Figure 7—figure supplement 2D*). In the naive states, mTORC1 primarily regulates keratinocyte proliferation, whereas mTORC2 mainly involved in the keratinocyte differentiation through Sema4A-related signaling pathways. Conversely, in the psoriatic dermatitis state, rapamycin downregulated both keratinocyte differentiation and proliferation markers. The observed similarities in *Il17a* expression following treatment with rapamycin and JR-AB2-011, regardless of additional IMQ treatment, suggest that *Il17a* production is not significantly dependent on Sema4A-related mTOR signaling.

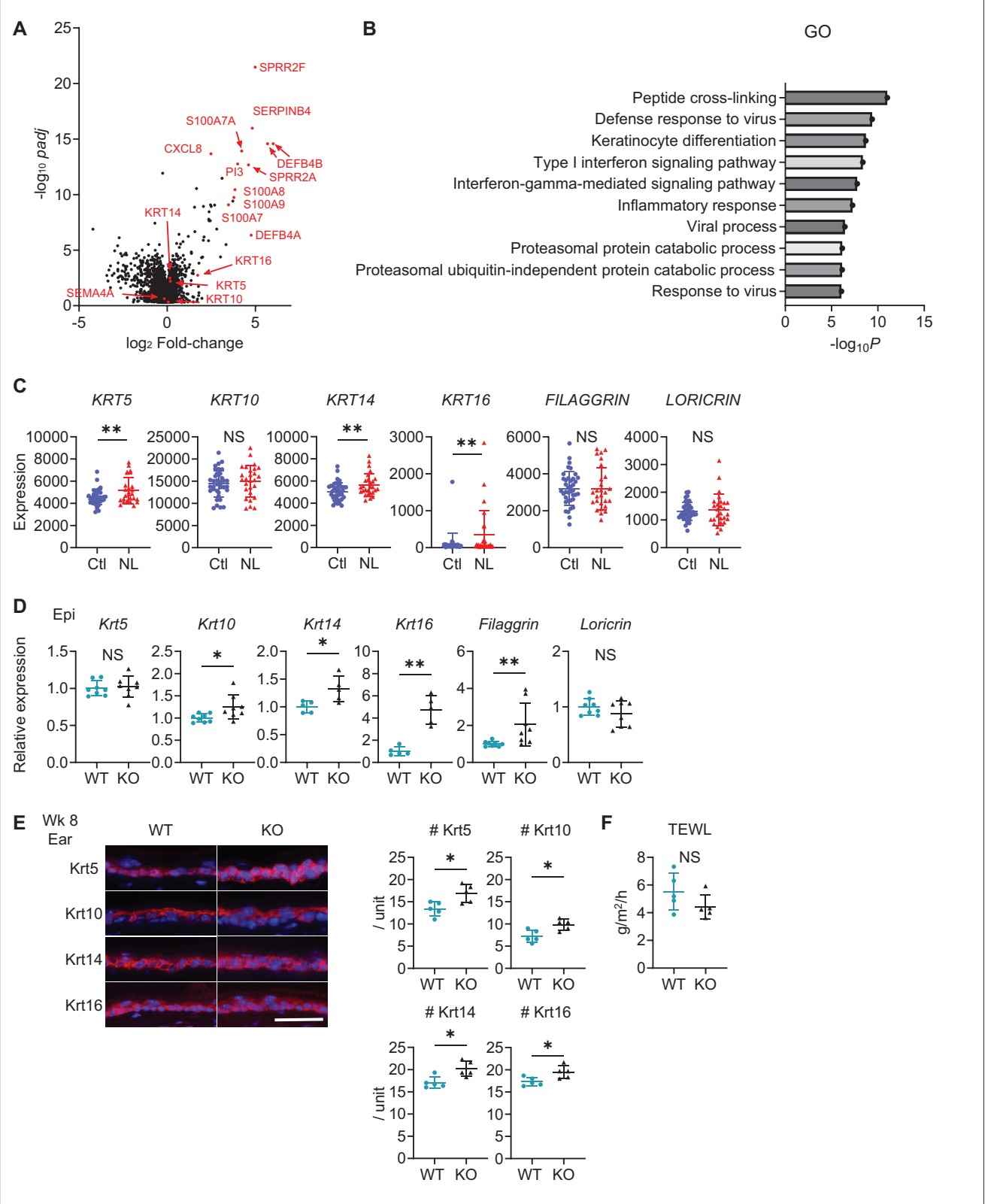

**Figure 5.** Sema4A knockout (KO) skin shares the features of human psoriatic non-lesions (NL). (**A, B**) The volcano plot (**A**) and Gene Ontology (GO) analysis (**B**), generated from RNA-sequencing data (GSE121212) using RaNAseq, display changes in gene expression in psoriatic NL compared to Ctl. (**C**) The difference in the expression of epidermal differentiation markers between Ctl and NL (n=38 for Ctl, n=27 for NL) was calculated with the transcripts per million values. **padj<0.01. NS, not significant. The error bars represent the standard deviation. (**D**) Relative gene expression of

*Figure 5 continued on next page*

*Figure 5 continued*

epidermal differentiation markers between wk 8 Epi of wild-type (WT) mice and KO mice (n=5 for *Krt14* and *Krt16*, n=8 for *Krt5*, *Krt10*, *Filaggrin,* and *Loricrin*). (**E**) Left: Representative immunofluorescence pictures of Krt5, Krt10, Krt14, and Krt16 (red) overlapped with DAPI. Scale bar = 50 μm. Right: Accumulated graphs showing the numbers of Krt5, Krt10, Krt14, and Krt16 positive cells per 100 μm width (n=5 per group) of wk 8 ear (right). Each dot represents the average from 5 unit areas per sample. (**F**) Transepidermal water loss (TEWL) in back skin of WT mice and KO mice at wk 8 (n=5 per group). (**D–F**) *p<0.05, **p<0.01. NS, not significant. The error bars represent the standard deviation.

The online version of this article includes the following source data and figure supplement(s) for figure 5:

**Source data 1.** Excel file containing quantitative data for *Figure 5*.

**Figure supplement 1.** The epidermis of psoriatic non-lesion is thicker than that of control skin.

**Figure supplement 1—source data 1.** Excel file containing quantitative data for *Figure 5—figure supplement 1*.

**Figure supplement 2.** Upregulation of cytokeratin expression related to psoriasis is not detected at birth in Sema4A knockout (KO) mice.

**Figure supplement 2—source data 1.** Excel file containing quantitative data for *Figure 5—figure supplement 2*.

## Discussion

Recent studies have highlighted the significance of IL-17A-producing $T_{RM}$ in psoriatic NL (*Cheuk et al., 2014*; *Gallais Sérézal et al., 2018*; *Vo et al., 2019*; *Vu et al., 2021*). This condition is also characterized by epidermal thickening with elevated CCL20 expression (*Gallais Sérézal et al., 2019*) and a distinct cytokeratin expression pattern, marked by increased levels of Krt5, Krt14, and Krt16 (*Tsoi et al., 2019*; *Figure 5C*). Here, we identified that murine Sema4A deficiency could induce these features of psoriatic NL.

Sema4A expression in epidermis, but not in dermis, was diminished in psoriatic L and NL compared to Ctl. While Sema4A expression on blood immune cells was upregulated in psoriasis, its serum levels were comparable between Ctl and psoriasis. Despite the reported involvement of increased Sema4A expression on immune cells in the pathogenesis of multiple sclerosis (*Koda et al., 2020*; *Nakatsuji et al., 2012*), our results suggest that the diminished expression of Sema4A in skin-constructing cells plays a more prominent role in the pathogenesis of psoriasis than its increased expression on immune cells.

In our murine model, regardless of IMQ application, IL-17A-producing T cells were increased in Sema4A KO skin although their frequency was comparable in dLN of WT mice and Sema4A KO mice. Additionally, the potential for in vitro T17 differentiation did not differ between T cells from WT mice and Sema4A KO mice. These findings suggest that the absence of Sema4A in the skin microenvironment plays a crucial role in the localized upregulation of IL-17A-producing T cells in Sema4A KO mice.

It is well documented that upregulated mTORC1 signaling promotes the pathogenesis of psoriasis (*Buerger, 2018*; *Karagianni et al., 2022*; *Ruf et al., 2014*), and that rapamycin can partially ameliorate the disease activity in psoriatic subjects and murine models (*Bürger et al., 2017*; *Gao and Si, 2018*; *Reitamo et al., 2001*). Our results using Sema4A KO mice revealed that inhibition of mTORC1 leads to the downregulation of *Krt16*, and inhibition of mTORC2 leads to the downregulation of undifferentiated keratinocytes, while the Sema4A KO epidermal thickening and the upregulated IL-17A signaling are not reversed by mTOR blockade in the naïve state. It is plausible that the downregulation of Sema4A can lead to the upregulation of mTORC1 and mTORC2 signaling in keratinocytes, and the augmented signaling leads to the psoriatic profile of proliferation and differentiation of keratinocytes, which is part of the psoriatic NL disposition.

This study has limitations. Sema4A expression in skin cells other than keratinocytes was not thoroughly investigated. However, since single-cell RNA-sequencing has shown that Sema4A is predominantly expressed in keratinocytes, dendritic cells, and macrophages, with a notable reduction in keratinocytes in psoriasis, we infer that keratinocytes are the primary cells responsible for the psoriatic features resulting from Sema4A downregulation. We were not able to determine whether Sema4A functions as a ligand or a receptor in epidermis (*Ito and Kumanogoh, 2016*; *Kumanogoh et al., 2002*; *Lu et al., 2018*; *Sun et al., 2017*) in this study, either. We were not able to reveal how Sema4A expression is regulated. Although we showed that downregulation of Sema4A is related to the abnormal cytokeratin expression observed in psoriasis, we could not determine the relationships between Sema4A expression and the essential molecules upregulated in psoriatic keratinocytes. While both mTORC1 and mTORC2 signals are upregulated in Sema4A KO epidermis, we were not able to confirm mTORC2 signaling from human skin due to technical limitation and sample limitation,

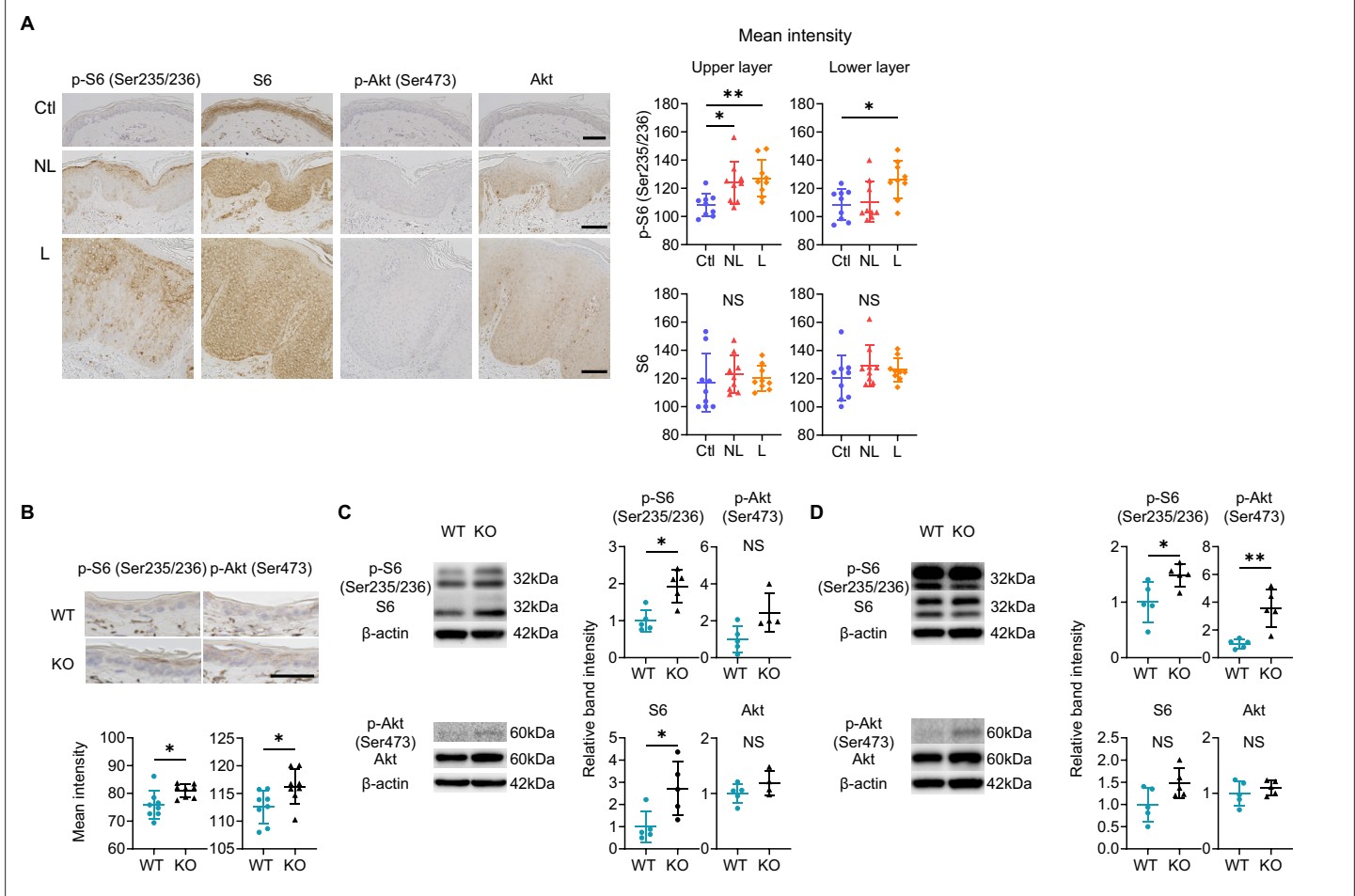

**Figure 6.** mTOR signaling is upregulated in the epidermis of psoriatic non-lesions (NL) and Sema4A knockout (KO) mice. (**A**) Representative results of immunohistochemistry displaying cells positive for phospho-S6 (p-S6) (Ser235/236), S6, phospho-Akt (p-Akt) (Ser473), and Akt in Ctl, NL, and L. The graphs of accumulated data show the mean intensity of p-S6 and S6 in the upper and lower epidermal layers (n=9 per group). Scale bar = 100 μm. Each dot represents the average mean intensity from 5 unit areas per sample. (**B**) The mean intensity of p-S6 (Ser235/236) and p-Akt (Ser473), detected by immunohistochemistry in the epidermis of wild-type (WT) mice and KO mice, were analyzed. Scale bar = 50 μm. Each dot represents the average intensity from 5 unit areas per sample (n=8 per group). (**C, D**) Immunoblotting of p-S6 (Ser235/236), S6, p-Akt (Ser473), and Akt in tissue lysates from epidermis without treatment (**C**) and with imiquimod (IMQ) treatment for consecutive 4 days (**D**) (n=5 per group, except for p-Akt and Akt in **C**, for which n=4). (**A–D**) *p<0.05, **p<0.01. NS, not significant. The error bars represent the standard deviation.

The online version of this article includes the following source data for figure 6:

**Source data 1.** Excel file containing quantitative data for *Figure 6*.

**Source data 2.** PDF file containing original western blots for *Figure 6C and D*.

**Source data 3.** Original JPG files for western blot analysis displayed in *Figure 6C and D*.

while the results from augmented mTORC2 signal in Sema4A KO mice and the normalization of *Krt5* and *Krt14* by mTORC2 inhibitor imply the involvement of mTORC2 signal in psoriatic epidermis. The role of Sema4A other than mTOR signaling, which may be involved in the regulation of T17 induction in skin, is not discussed, either. These limitations should be overcome in the near future.

In summary, epidermal Sema4A downregulation can reflect the psoriatic non-lesional features. Thus, targeting the downregulated Sema4A and upregulated mTOR signaling in psoriatic epidermis can be a promising strategy for psoriasis therapy and modification of psoriatic diathesis in NL for the prevention of disease development.

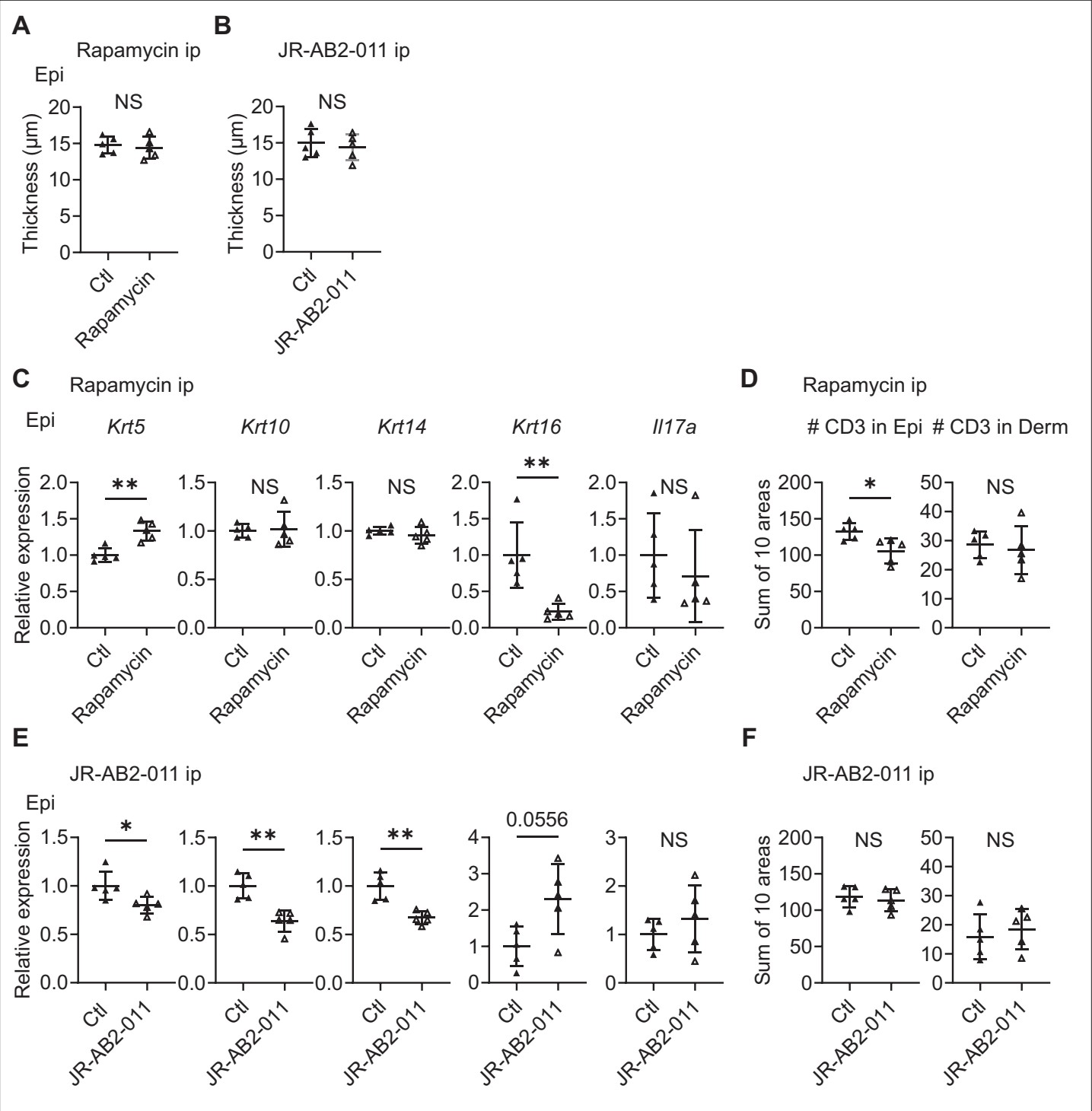

**Figure 7.** Inhibitors of mTOR signaling modulate the expression of cytokeratins in Sema4A knockout (KO) mice. (**A, B**) Epidermal thickness of Sema4A KO mice treated intraperitoneally with vehicle (Ctl) or rapamycin (**A**), and Ctl or JR-AB2-011 (**B**) (n=5 per group). (**C, D**) Relative expression of keratinocyte differentiation markers and *Il17a* in Sema4A KO Epi (**C**), and the number of T cells in Epi and Derm under Ctl or rapamycin (**D**) (n=5 per group). (**E, F**) Relative expression of keratinocyte differentiation markers and *Il17a* in Sema4A KO Epi (**E**), and the number of T cells in Epi and Derm under Ctl or JR-AB2-011 (**F**) (n=5 per group). (**D** and **F**) Each dot represents the sum of numbers from 10 unit areas across 3 specimens. (**A–F**) *p<0.05, **p<0.01. NS, not significant. The error bars represent the standard deviation.

The online version of this article includes the following source data and figure supplement(s) for figure 7:

**Source data 1.** Excel file containing quantitative data for *Figure 7*.

*Figure 7 continued on next page*

*Figure 7 continued*

**Figure supplement 1.** Topical application of rapamycin gel yields partially similar results to intraperitoneal treatment.

**Figure supplement 1—source data 1.** Excel file containing quantitative data for *Figure 7—figure supplement 1*.

**Figure supplement 2.** Rapamycin treatment reduced the epidermal swelling observed in imiquimod (IMQ)-treated Sema4A knockout (KO) mice.

**Figure supplement 2—source data 1.** Excel file containing quantitative data for *Figure 7—figure supplement 2*.

## Materials and methods

### Human sample collection

Psoriatic L and NL skin specimens were acquired from 17 psoriasis patients. Ctl specimens were obtained from 19 subjects who underwent tumor resection or reconstructive surgery. For epidermal-dermal separation, specimens were incubated overnight at 4°C with 2.5 mg/mL dispase II (Wako, Osaka, Japan) in IMDM (Wako). Blood samples were collected from 73 psoriasis and 33 Ctl. In addition to ELISA (*Nakatsuji et al., 2012*), peripheral blood mononuclear cells were isolated using Ficoll-Paque PLUS density gradient media (Cytiva, Tokyo, Japan). Patient details are provided in *Supplementary file 1*.

### Mice

C57BL/6J WT mice were procured from CLEA Japan (Tokyo, Japan). Sema4A KO mice with C57BL/6J background were generated as previously described (*Kumanogoh et al., 2005*). We examined female mice in order to reduce the result variation. Neonatal mice and female mice aged 8–12 weeks, maintained under specific pathogen-free conditions, were used.

To develop psoriasis-like dermatitis, 10 mg of 5% IMQ cream (Mochida, Tokyo, Japan) was applied to both ears for 4 consecutive days. To induce IL-23-mediated psoriasis-like dermatitis, 20 µL of phosphate-buffered saline containing 500 ng of recombinant mouse IL-23 (BioLegend, San Diego, CA, USA) was injected intradermally into both ears of anesthetized mice using a 29-gauge needle for 4 consecutive days. Ear thickness was measured using a thickness gauge (Peacock, Tokyo, Japan).

For bone marrow reconstitution, $2 \times 10^5$ bone marrow cells obtained from the indicated strains were intravenously transferred to recipient mice that had received a single 10 Gy irradiation. The reconstituted mice were subjected to the experiments in 8 weeks.

In the specified experiments, skin specimens were separated into epidermis and dermis by 5 mg/mL dispase (Wako) in IMDM for 30 min at 37°C.

Transepidermal water loss measurements were performed on the back skin of mice at week 8 using a Tewameter (Courage and Khazaka Electronic GmbH, Cologne, Germany), according to the manufacturer's instructions.

### Immunofluorescence and immunohistochemistry

Specimens were fixed in 4% paraformaldehyde phosphate buffer solution (Wako), embedded in paraffin, and sliced into 3 µm thickness on glass slides. After deparaffinization and rehydration, antigen retrieval was performed using citrate buffer (pH 6.0, Nacalai Tesque, Kyoto, Japan) or TE buffer (pH 9.0, Nacalai Tesque).

For immunohistochemistry, samples were incubated with 3% $H_2O_2$ (Wako) for 5 min. After blocking (Agilent, Santa Clara, CA, USA), the specimens were incubated with the indicated primary antibodies under specified conditions (*Supplementary file 2*). The specimens were applied with the Dako REAL EnVision Detection System, Peroxidase/DAB, Rabbit/Mouse, HRP kit (Agilent) and counterstained with hematoxylin (Wako). For immunofluorescence, the blocked specimens were incubated with the indicated primary antibodies followed by the secondary antibodies (*Supplementary file 2*). Mounting medium with DAPI (Vector Laboratories, Burlingame, CA, USA) was used. Slides were observed using a fluorescence microscope (BZ-X700, Keyence, Osaka, Japan). The staining intensity was measured using Fiji software (ImageJ, National Institutes of Health, Bethesda, MD, USA) over lengths of 200 µm in human samples and 100 µm in murine samples. This measurement was taken from 5 areas of each specimen, and the average score is presented. The average thickness of epidermis and dermis from 10 spots is presented. The number of epidermal cells positive for the indicated cytokeratins was counted per 100 µm width, with the average number from 5 areas being presented. Additionally, the

numbers of CD3-positive cells in murine ears were counted across 10 fields of 400 μm width, with the total sum being presented.

## Quantitative reverse transcription polymerase chain reaction

Total RNA was extracted from homogenized skin tissue using Direct-zol RNA MiniPrep Kit (Zymo Research, Irvine, CA, USA). cDNA was synthesized by High-Capacity RNA-to-cDNA Kit (Thermo Fisher Scientific, Waltham, MA, USA). qPCR was performed using TB Green Premix Ex Taq II (Takara Bio, Shiga, Japan) on a ViiA 7 Real-Time PCR System (Thermo Fisher Scientific). The primers are listed in *Supplementary file 3*. All samples were run in triplicate, and the median CT value was calculated. Relative gene expression levels were normalized to the housekeeping gene *GAPDH* using the ΔΔCT technique.

## Flow cytometry analysis

Single-cell suspensions from murine skin-dLN and spleen were prepared by grinding, filtering, and lysing red blood cells (BioLegend). Skin specimens were minced and digested with 3 mg/mL collagenase type III (Worthington Biochemical Corporation, Lakewood, NJ, USA) in RPMI 1640 medium (Wako) at 37°C for 10 min for epidermis, and 30 min for dermis or whole skin. Cells were surface-stained with directly conjugated monoclonal antibodies (*Supplementary file 4*). Dead cells were identified using LIVE/DEAD Fixable Dead Cell Stain Kit (Thermo Fisher Scientific). To evaluate cytokine production, cells were stimulated with Phorbol 12-Myristate 13-Acetate (PMA; 50 ng/mL, Wako) and ionomycin (1000 ng/mL, Wako), plus BD Golgiplug (BD Biosciences, San Jose, CA, USA) for 4 hr before surface staining. Fixation, permeabilization, and intracellular cytokine staining were performed using BD Cytofix/Cytoperm Fixation/Permeabilization Kit (BD Biosciences) according to the manufacturer's protocol. In specified experiments, the numbers of each cell subset per ear were estimated using CountBright Absolute Counting Beads (Thermo Fisher Scientific). Sample analysis was conducted using BD FACSCanto II (BD Biosciences), and data were analyzed with Kaluza software (Beckman Coulter, Brea, CA, USA).

## In vitro Th17 differentiation

Murine splenic T cells were isolated using Pan T Cell Isolation Kit II (Miltenyi Biotec, Bergisch Gladbach, Germany). Two hundred thousand T cells per well were cultured in 96-well plates in the presence of T Cell Activation/Expansion Kit (Miltenyi Biotec) for 2 weeks. The medium was supplemented twice per week with the following recombinant cytokines: mouse recombinant IL-6 (20 ng/mL), IL-1β (20 ng/mL), and IL-23 (40 ng/mL) for the IL-23-dependent Th17 cell condition; IL-6 (20 ng/mL) and TGFβ1 (3 ng/mL) for the IL-23-independent Th17 cell condition; or IL-23 (40 ng/mL) for the IL-23 only Th17 cell condition. The cytokines listed were purchased from BioLegend (*Supplementary file 5*). Afterward, the cultured cells were processed for flow cytometry analysis.

## Western blotting

Murine epidermis was lysed using RIPA buffer (Wako) containing phosphatase and protease inhibitor cocktail (Nacalai Tesque). Protein lysates were separated by 10% SuperSep Ace (Wako), transferred onto polyvinylidene difluoride membranes (0.45 μm, Merck, Darmstadt, Germany) by Trans-Blot Turbo Transfer System (Bio-Rad, Hercules, CA, USA). The membrane was blocked with 5% bovine serum albumin and subjected to immunoblotting targeting the indicated proteins overnight 4°C, followed by the application of HRP-conjugated secondary antibody for 1 hr at room temperature (*Supplementary file 2*). WB stripping solution (Nacalai Tesque) was used to remove the antibodies for further evaluation.

## Inhibition of mTOR

Rapamycin (4 mg/kg, Sanxin Chempharma, Hebei, China), JR-AB2-011 (400 μg/kg, MedChemExpress, Monmouth Junction, NJ, USA) and vehicle were applied intraperitoneally to mice once daily for 14 consecutive days. Rapamycin and JR-AB2-011 were dissolved in 100% DMSO and diluted with 40% Polyethylene Glycol 300 (Wako), 5% Tween-80 (Sigma-Aldrich, St. Louis, MO, USA) and 45% saline in sequence. For topical application, 0.2% rapamycin gel and vehicle gel were prepared by the pharmaceutical department as previously described (*Wataya-Kaneda et al., 2017*). Rapamycin gel

was applied to the left ear, and vehicle gel to the right ear (10 mg/ear) of Sema4A KO, once daily for 14 consecutive days. To analyze the preventive effectiveness of rapamycin in an IMQ-induced murine model of psoriatic dermatitis, Sema4A KO mice were administered either vehicle or rapamycin intraperitoneally from day 0 to day 17, and IMQ was topically applied to both ears for 4 days starting on day 14. Then, on day 18, ears were collected for further analysis.

## Data processing of single-cell RNA-sequencing and bulk RNA-sequencing

The raw count matrix data from the previously reported single-cell RNA-sequencing data from GSE220116 (*Kim et al., 2023*) were imported into Scanpy (1.9.6) using Python for further analyses. For each sample, cells and genes meeting the following criteria were excluded: cells expressing over 200 genes (sc.pp.filter_cells), cells with a high proportion of mitochondrial genes (>5%), and genes expressed in fewer than 3 cells (sc.pp.filter_genes). Counts were normalized using sc.pp.normalize_per_cell, logarithmized (sc.pp.log1p), and scaled (sc.pp.scale). Highly variable genes were selected using sc.pp.filter_genes_dispersion with the options min_mean = 0.0125, max_mean = 2.5, and min_disp = 0.7.

Principal component analysis was conducted with sc.pp.pca, selecting the 1st to 50th principal components for embedding and clustering. Neighbors were calculated with batch effect correction by BBKNN (*Polański et al., 2020*). Embedding was performed by sc.tl.umap, and cells were clustered using sc.tl.leiden. Some cells were further subclassified in a similar manner. The data was integrated into an h5ad file, which can be visualized in Cellxgene VIP (*Li et al., 2022*). We then performed differential analysis between two groups of cells to identify differential expressed genes using Welch's t-test. Multiple comparisons were controlled using the Benjamini-Hochberg procedure, with the false discovery rate set at 0.05 and significance defined as padj<0.05.

Bulk RNA-sequencing data from GSE121212 (*Tsoi et al., 2019*) were re-analyzed using RaNAseq (*Prieto and Barrios, 2019*) for differential expression in Ctl versus psoriatic NL. Our analysis included normalization, differential gene expression (defining significance as padj<0.05), and focused on Gene Ontology biological process analysis. The gene expression was calculated with the transcripts per million values.

## Statistical analysis

GraphPad Prism 10 software (GraphPad Software, La Jolla, CA, USA) was used for all statistical analyses except for RNA-sequencing. Mann-Whitney test was used for two-group comparisons, while Kruskal-Wallis test followed by Dunn's multiple comparisons test was applied for comparison among three or more groups. Statistical significance was defined as p<0.05 (*), p<0.01 (**), and p<0.001 (***). In the analysis of RNA-sequencing data, statistical significance was defined as padj<0.01 (**) and padj<0.001 (***).

## Acknowledgements

We express our gratitude to all patients for their participation in this study.

## Additional information

### Competing interests

Shuichi Nakai, Shoichi Matsuda: affiliated with Maruho Co. as an employee, but has declared no conflicts of interest related to this research. The other authors declare that no competing interests exist.

### Funding

| Funder | Grant reference number | Author |
| --- | --- | --- |
| Japan Society for the Promotion of Science | 22KJ2071 | Miki Kume |

| Funder | Grant reference number | Author |
| --- | --- | --- |
| Japan Society for the Promotion of Science | 16K19705 | Rei Watanabe |
| Maruho | collaborative research grant | Rei Watanabe |

The funders had no role in study design, data collection and interpretation, or the decision to submit the work for publication.

## Author contributions

Miki Kume, Conceptualization, Resources, Data curation, Software, Formal analysis, Funding acquisition, Validation, Investigation, Visualization, Methodology, Writing - original draft, Project administration, Writing – review and editing; Hanako Koguchi-Yoshioka, Conceptualization, Data curation, Software, Formal analysis, Validation, Investigation, Visualization, Methodology, Writing - original draft, Project administration, Writing – review and editing; Shuichi Nakai, Yutaka Matsumura, Shoichi Matsuda, Data curation, Validation, Writing – review and editing; Atsushi Tanemura, Resources, Data curation, Validation, Writing – review and editing; Kazunori Yokoi, Data curation, Software, Validation, Writing – review and editing; Yuumi Nakamura, Atsushi Kumanogoh, Supervision, Validation, Writing – review and editing; Naoya Otani, Mifue Taminato, Resources, Validation, Writing – review and editing; Koichi Tomita, Tateki Kubo, Resources, Supervision, Validation, Writing – review and editing; Mari Wataya-Kaneda, Supervision, Validation, Writing – review and editing, Provision of reagents; Manabu Fujimoto, Conceptualization, Supervision, Validation, Project administration, Writing – review and editing; Rei Watanabe, Conceptualization, Resources, Data curation, Software, Formal analysis, Supervision, Funding acquisition, Validation, Investigation, Visualization, Methodology, Writing - original draft, Project administration, Writing – review and editing

## Author ORCIDs

Miki Kume https://orcid.org/0000-0001-7632-6072
Shuichi Nakai https://orcid.org/0000-0002-6884-7705
Rei Watanabe https://orcid.org/0000-0001-8254-9176

## Ethics

All experiments involving human specimens were in accordance with the Declaration of Helsinki and were approved by the Institutional Review Board in Osaka University Hospital (20158-6). Written informed consent was obtained from all participants.

All murine experiments were approved by Osaka University Animal Experiment Committee (J007591-013) and all procedures were conducted in compliance with the Guidelines for Animal Experimentation established by Japanese Association for Laboratory Animal Science.

Reviewer #1 (Public review): https://doi.org/10.7554/eLife.97654.3.sa1
Reviewer #2 (Public review): https://doi.org/10.7554/eLife.97654.3.sa2
Author response https://doi.org/10.7554/eLife.97654.3.sa3

# Additional files

## Supplementary files

Supplementary file 1. Patient information. This table provides patient information used in this study. Psoriasis severity was defined by the total body surface area (BSA) affected: <3% BSA for mild, 3–10% BSA for moderate, and >10% BSA for severe disease.

Supplementary file 2. Antibodies used for immunohistochemical, immunofluorescence, and western blot analyses.

Supplementary file 3. Primer sequences used for real-time quantitative PCR in experiments with human and murine samples.

Supplementary file 4. Antibodies used for flow cytometry analysis.

Supplementary file 5. Mouse recombinant cytokines.

MDAR checklist

### Data availability

The single-cell RNA-sequencing datasets generated by *Kim et al., 2023* and *Tsoi et al., 2019* used in this study are available in the NCBI Gene Expression Omnibus under accession codes GSE220116 and GSE121212, respectively. Values for all data points in graphs are reported in source data.

The following previously published datasets were used:

| Author(s) | Year | Dataset title | Dataset URL | Database and Identifier |
|---|---|---|---|---|
| Kim J, Krueger JG | 2023 | Single-cell transcriptomics suggest distinct upstream drivers of IL-17A/F in hidradenitis versus psoriasis | https://www.ncbi.nlm.nih.gov/geo/query/acc.cgi?acc=GSE220116 | NCBI Gene Expression Omnibus, GSE220116 |
| Weidinger S, Rodriguez E, Tsoi LC, Gudjonsson J | 2019 | Atopic Dermatitis, Psoriasis and healthy control RNA-seq cohort | https://www.ncbi.nlm.nih.gov/geo/query/acc.cgi?acc=GSE121212 | NCBI Gene Expression Omnibus, GSE121212 |

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

# Appendix 1

## Appendix 1—key resources table

| Reagent type (species) or resource | Designation | Source or reference | Identifiers | Additional information |
|---|---|---|---|---|
| Strain, strain background (*Mus musculus*) | C57BL/6J WT mice | CLEA Japan | | |
| Strain, strain background (*M. musculus*) | Semaphorin 4A knockout mice | Dr. Atsushi Kumanogoh (Osaka University, Osaka, Japan) | | |
| Biological sample (*Homo sapiens*) | Skin specimens from 17 psoriasis patients | Osaka University | | |
| Biological sample (*H. sapiens*) | Skin specimens from 19 subjects who underwent tumor resection or reconstructive surgery | Osaka University | | |
| Biological sample (*H. sapiens*) | Blood samples from 73 psoriasis patients | Osaka University | | |
| Biological sample (*H. sapiens*) | Blood samples from 33 Ctl | Osaka University | | |
| Antibody | Sema4A (Rabbit polyclonal) | Abcam | Cat# Ab70178; RRID:AB_1270611 | IHC (1:100) |
| Antibody | Phospho-S6 (Ser235/236) (Rabbit monoclonal) | Cell Signaling Technology | Cat# 4858; RRID:AB_916156 | IHC (1:400), WB (1:2000) |
| Antibody | S6 (Rabbit monoclonal) | Cell Signaling Technology | Cat# 2217; RRID:AB_331355 | IHC (1:100), WB (1:1000) |
| Antibody | Phospho-Akt (Ser473) (Rabbit monoclonal) | Cell Signaling Technology | Cat# 4060; RRID:AB_2315049 | IHC (1:50), WB (1:1000) |
| Antibody | Akt (Rabbit monoclonal) | Cell Signaling Technology | Cat# 4691; RRID:AB_915783 | IHC (1:300), WB (1:1000) |
| Antibody | Keratin 5 (Rabbit polyclonal) | BioLegend | Cat# 905503; RRID:AB_2734679 | IF (1:800) |
| Antibody | Keratin 10 (Rabbit polyclonal) | BioLegend | Cat# 905403; RRID:AB_2749902 | IF (1:400) |
| Antibody | Keratin 14 (Rabbit polyclonal) | BioLegend | Cat# 905303; RRID:AB_2734678 | IF (1:400) |
| Antibody | Cytokeratin 16 (Rabbit monoclonal) | Invitrogen | Cat# MA5-42892; RRID:AB_2912033 | IF (1:100) |
| Antibody | CD3 (Rat monoclonal) | Bio-Rad | Cat# MCA1477; RRID:AB_321245 | IF (1:100) |
| Antibody | Rabbit IgG H&L (Alexa Fluor 555) (Donkey polyclonal) | Abcam | Cat# ab150074; RRID:AB_2636997 | IF (1:1000) |
| Antibody | Rat IgG H&L (Alexa Fluor 555) (Donkey polyclonal) | Abcam | Cat# ab150154; RRID:AB_2813834 | IF (1:1000) |
| Antibody | β-Actin (Mouse monoclonal) | Sigma-Aldrich | Cat# A5441; RRID:AB_476744 | WB (1:5000) |
| Antibody | Anti-Mouse IgG, HRP-Linked Whole Ab (Sheep monoclonal secondary) | Cytiva | Cat# NA931; RRID:AB_772210 | WB (1:10,000) |
| Antibody | Anti-Rabbit IgG, HRP-Linked Whole Ab (Donkey polyclonal secondary) | Cytiva | Cat# NA934; RRID:AB_772206 | WB (1:10,000) |
| Antibody | CD3 (Mouse monoclonal) | BioLegend | Cat# 317335; RRID:AB_2561627 | FCM (1:100) |
| Antibody | CD4 (Mouse monoclonal) | BioLegend | Cat# 300512; RRID:AB_314080 | FCM (1:100) |
| Antibody | CD8a (Mouse monoclonal) | eBioscience | Cat# 47-0088-42; RRID:AB_1272046 | FCM (1:100) |
| Antibody | CD11c (Mouse monoclonal) | BioLegend | Cat# 301628; RRID:AB_11203895 | FCM (1:100) |

*Appendix 1 Continued on next page*

*Appendix 1 Continued*

| Reagent type (species) or resource | Designation | Source or reference | Identifiers | Additional information |
|---|---|---|---|---|
| Antibody | SEMA4A (Mouse monoclonal) | BioLegend | Cat# 148404; RRID:AB_2565287 | FCM (1:100) |
| Antibody | CD3ε (Armenian Hamster monoclonal) | BioLegend | Cat# 100328; RRID:AB_893318 | FCM Skin specimens (1:20), Others (1:100) |
| Antibody | CD4 (Rat monoclonal) | BioLegend | Cat# 100406; RRID:AB_312691 | FCM (1:100) |
| Antibody | CD8a (Rat monoclonal) | BioLegend | Cat# 100714; RRID:AB_312753 | FCM (1:100) |
| Antibody | CD16/32 (Rat monoclonal) | BioLegend | Cat# 101301; RRID:AB_312800 | FCM (1:100) |
| Antibody | CD69 (Armenian Hamster monoclonal) | BioLegend | Cat# 104514; RRID:AB_492843 | FCM (1:10) |
| Antibody | CD103 (Armenian Hamster monoclonal) | BioLegend | Cat# 121422; RRID:AB_2562901 | FCM (1:100) |
| Antibody | TCR Vγ2 (Armenian Hamster monoclonal) | BioLegend | Cat# 137705; RRID:AB_10643997 | FCM (1:100) |
| Antibody | TCR Vγ3 (Syrian Hamster monoclonal) | BD Biosciences | Cat# 743241; RRID:AB_2741371 | FCM (1:100) |
| Antibody | TCRγδ (Armenian Hamster monoclonal) | BioLegend | Cat# 118124; RRID:AB_11204423 | FCM (1:100) |
| Antibody | IFNγ (Rat monoclonal) | BioLegend | Cat# 505813; RRID:AB_493312 | FCM (1:40) |
| Antibody | IL-4 (Rat monoclonal) | BD Biosciences | Cat# 562915; RRID:AB_2737889 | FCM (1:40) |
| Antibody | IL-17A (Rat monoclonal) | BioLegend | Cat# 506925; RRID:AB_10900442 | FCM (1:40) |
| Sequence-based reagent | Human GAPDH_F | This paper | PCR primers | GTCTCCTCTGACTTCAACAGCG |
| Sequence-based reagent | Human GAPDH_R | This paper | PCR primers | ACCACCCTGTTGCTGTAGCCAA |
| Sequence-based reagent | Human SEMA4A_F | *Carvalheiro et al., 2019* | PCR primers | TCTGCTCCTGAGTGGTGATG |
| Sequence-based reagent | Human SEMA4A_R | *Carvalheiro et al., 2019* | PCR primers | AAACCAGGACACGGATGAAG |
| Peptide, recombinant protein | Recombinant Mouse IL-1β (carrier-free) | BioLegend | Cat# 575102 | |
| Peptide, recombinant protein | Recombinant Mouse IL-6 (carrier-free) | BioLegend | Cat# 575702 | |
| Peptide, recombinant protein | Recombinant Mouse IL-23 (carrier-free) | BioLegend | Cat# 589006 | |
| Peptide, recombinant protein | Recombinant Mouse TGF-β1 (carrier-free) | BioLegend | Cat# 763102 | |
| Commercial assay or kit | BD Cytofix/Cytoperm Fixation/Permeabilization Kit | BD Biosciences | Cat# 554714 | |
| Commercial assay or kit | Dako REAL EnVision Detection System, Peroxidase/DAB, Rabbit/Mouse, HRP kit | Agilent | Cat# K5007; RRID:AB_2888627 | |
| Commercial assay or kit | Direct-zol RNA Miniprep Kits | Zymo Research | Cat# R2050 | |
| Commercial assay or kit | High-Capacity RNA-to-cDNA Kit | Thermo Fisher scientific | Cat# 4387406 | |
| Commercial assay or kit | LIVE/DEAD Fixable Dead Cell Stain Kit | Thermo Fisher Scientific | Cat# L34965 | |

*Appendix 1 Continued*

| Reagent type (species) or resource | Designation | Source or reference | Identifiers | Additional information |
|---|---|---|---|---|
| Commercial assay or kit | Pan T Cell Isolation Kit II, mouse | Miltenyi Biotec | Cat# 130-095-130 | |
| Commercial assay or kit | T Cell Activation/Expansion Kit, mouse | Miltenyi Biotec | Cat# 130-093-627 | |
| Commercial assay or kit | TB Green Premix Ex Taq II (Tli RNaseH Plus) | Takara Bio | Cat# RR820A | |
| Chemical compound, drug | 5% imiquimod cream | Mochida | Global Trade Item Number: 224130002 | |
| Chemical compound, drug | BD Golgiplug | BD Biosciences | Cat# 555029 | |
| Chemical compound, drug | Collagenase type III | Worthington Biochemical Corporation | Cat# LS004183 | |
| Chemical compound, drug | CountBright Absolute Counting Beads, for flow cytometry | Thermo Fisher Scientific | Cat# C36950 | |
| Chemical compound, drug | Dispase II | Wako | Cat# 383-02281 | |
| Chemical compound, drug | Ionomycin | Wako | Cat# 095-05831 | |
| Chemical compound, drug | JR-AB2-011 | MedChemExpress | Cat# HY-122022 | |
| Chemical compound, drug | Mounting medium with DAPI | Vector Laboratories | Cat# H-1200; RRID:AB_2336790 | |
| Chemical compound, drug | Phorbol 12-Myristate 13-Acetate | Wako | Cat# 162-23591 | |
| Chemical compound, drug | Phosphatase Inhibitor Cocktail (100×) | Nacalai Tesque | Cat# 07574-61 | |
| Chemical compound, drug | Protease Inhibitor Cocktail for Use with Mammalian Cell and Tissue Extracts | Nacalai Tesque | Cat# 25955-11 | |
| Chemical compound, drug | Protein Block Serum-Free | Agilent | Cat# X0909 | |
| Chemical compound, drug | Rapamycin | Sanxin Chempharma | CAS# 53123-88-9 | |
| Chemical compound, drug | RBC Lysis Buffer (10×) | BioLegend | Cat# 420301 | |
| Chemical compound, drug | WB Stripping Solution | Nacalai Tesque | Cat# 05364-55 | |
| Software, algorithm | Cellxgene VIP | *Li et al., 2022* | | |
| Software, algorithm | GraphPad Prism 10 | GraphPad Software | RRID:SCR_002798 | |
| Software, algorithm | ImageJ | National Institutes of Health | RRID:SCR_003070 | |
| Software, algorithm | Kaluza | Beckman Coulter | RRID:SCR_016182 | |
| Software, algorithm | RaNAseq | https://ranaseq.eu/; *Prieto and Barrios, 2019* | | |

