## [Editor Report · eLife Assessment]

This paper advances an **important** new concept in psoriasis pathogenesis and implicates Sema4a as a homeostatic regulator that is highly epithelial-specific. The findings are **convincing** and lend support for the biology described here as a mechanism with therapeutic implications.

---

## [Referee Report · Reviewer #1 (Public review)]

Summary:

In this study, Kume et al examined the role of the protein Semaphorin 4a in steady state skin homeostasis and how this relates to skin changes seen in human psoriasis and imiquimod-induced psoriasis-like disease in mice. The authors found that human psoriatic skin has reduced expression of Sema4a in the epidermis. While Sema4a has been shown to drive inflammatory activation in different immune populations, this finding suggested Sema4a might be important for negatively regulating Th17 inflammation in the skin. The authors go on to show that Sema4a knockout mice have skin changes in key keratinocyte genes, increased gdT cells, and increased IL-17 similar to differences seen in non-lesional psoriatic skin, and that bone marrow chimera mice with WT immune cells and Sema4a KO stromal cells develop worse IMQ-induced psoriasis-like disease, further linking expression of Sema4a in the skin to maintaining skin homeostasis. The authors next studied downstream pathways that might mediate the homeostatic effects of Sema4a, focusing on mTOR given its known role in keratinocyte function. Like for the immune phenotypes, Sema4a KO mice had increased mTOR activation in the epidermis in a similar pattern to mTOR activation noted in non-lesional psoriatic skin. The authors next targeted the mTOR pathway and showed rapamycin could reverse some of the psoriasis-like skin changes in Sema4a KO mice, confirming the role of increased mTOR in contributing to the observed skin phenotype.

In the revised manuscript, the authors expand on the potential relevance to psoriasis by demonstrating similar findings in an IL-23-diriven model of skin inflammation, which is an orthogonal model of psoriasis to their original IMQ model. They also show that in addition to reversing steady state differences in skin thickness between Sema4a KO mice and WT mice, rapamycin improves metrics of disease in the IMQ model of psoriasis. These additional studies further bolster their conclusions that Sema4a may play a protective role in by preventing over-activation of mTOR in the skin in psoriasis.

Strengths:

The most interesting finding is the tissue-specific role for Sema4a, where it has previously been considered to play a mostly pro-inflammatory role in immune cells, this study shows that when expressed by keratinocytes, Sema4a plays a homeostatic role that when missing leads to development of psoriasis-like skin changes. This has important implications in terms of targeting Sema4a pharmacologically. It also may yield a novel mouse model to study mechanisms of psoriasis development in mice separate from the commonly used IMQ model. The included experiments are well-controlled and executed rigorously.

The new experiments provide additional data to support the conclusions through an orthogonal model of psoriasis and demonstrating rapamycin-induced reversal of changes in the IMQ disease model.

Weaknesses:

While the main weakness of these studies, lack of tissue-specific Sema4a knockout mice (e.g. in keratinocytes only), remains, generating these mice and performing the necessary experiments is beyond the scope of completing these particular studies. Similarly, it is understandable that additional bone marrow chimeras would be costly and labor intensive without adding much more in the absence of tissue-specific knockouts.

---

## [Referee Report · Reviewer #2 (Public review)]

Summary:

Kume et al. found for the first time that Semaphorin 4A (Sema4A) was downregulated in both mRNA and protein levels in L and NL keratinocytes of psoriasis patients compared to control keratinocytes. In peripheral blood, they found that Sema4A is not only expressed in keratinocytes but is also upregulated in hematopoietic cells such as lymphocytes and monocytes in the blood of psoriasis patients. They investigated how the down-regulation of Sema4A expression in psoriatic epidermal cells affects the immunological inflammation of psoriasis by using a psoriasis mice model in which Sema4A KO mice were treated with IMQ. Kume et al. hypothesized that down-regulation of Sema4A expression in keratinocytes might be responsible for the augmentation of psoriasis inflammation. Using bone marrow chimeric mice, Kume et al. showed that KO of Sema4A in non-hematopoietic cells was responsible for the enhanced inflammation in psoriasis. The expression of CCL20, TNF, IL-17, and mTOR was upregulated in the Sema4AKO epidermis compared to the WT epidermis, and the infiltration of IL-17-producing T cells was also enhanced.

Strengths:

Decreased Sema4A expression may be involved in psoriasis exacerbation through epidermal proliferation and enhanced infiltration of Th17 cells, which helps understand psoriasis immunopathogenesis.

Weaknesses:

The mechanism of decreased Sema4A expression in psoriasis is not clear, although this does not affect the strength of this research.

---

## [Author Response]

The following is the authors’ response to the original reviews.

**Public reviews:**

**Reviewer #1 (Public Review):**
Summary:In this study, Kume et al examined the role of the protein Semaphorin 4a in steady-state skin homeostasis and how this relates to skin changes seen in human psoriasis and imiquimod-induced psoriasis-like disease in mice. The authors found that human psoriatic skin has reduced expression of Sema4a in the epidermis. While Sema4a has been shown to drive inflammatory activation in different immune populations, this finding suggested Sema4a might be important for negatively regulating Th17 inflammation in the skin. The authors go on to show that Sema4a knockout mice have skin changes in key keratinocyte genes, increased gdT cells, and increased IL-17 similar to differences seen in non-lesional psoriatic skin, and that bone marrow chimera mice with WT immune cells and Sema4a KO stromal cells develop worse IMQ-induced psoriasis-like disease, further linking expression of Sema4a in the skin to maintaining skin homeostasis. The authors next studied downstream pathways that might mediate the homeostatic effects of Sema4a, focusing on mTOR given its known role in keratinocyte function. As with the immune phenotypes, Sema4a KO mice had increased mTOR activation in the epidermis in a similar pattern to mTOR activation noted in non-lesional psoriatic skin. The authors next targeted the mTOR pathway and showed rapamycin could reverse some of the psoriasis-like skin changes in Sema4a KO mice, confirming the role of increased mTOR in contributing to the observed skin phenotype.Strengths:The most interesting finding is the tissue-specific role for Sema4a, where it has previously been considered to play a mostly pro-inflammatory role in immune cells, this study shows that when expressed by keratinocytes, Sema4a plays a homeostatic role that when missing leads to the development of psoriasis-like skin changes. This has important implications in terms of targeting Sema4a pharmacologically. It also may yield a novel mouse model to study mechanisms of psoriasis development in mice separate from the commonly used IMQ model. The included experiments are well-controlled and executed rigorously.Weaknesses:A weakness of the study is the lack of tissue-specific Sema4a knockout mice (e.g. in keratinocytes only). The authors did use bone marrow chimeras, but only in one experiment. This work implies that psoriasis may represent a Sema4a-deficient state in the epidermal cells, while the same might not be true for immune cells. Indeed, in their analysis of non-lesional psoriasis skin, Sema4a was not significantly decreased compared to control skin, possibly due to compensatory increased Sema4a from other cell types. Unbiased RNA-seq of Sema4a KO mouse skin for comparison to non-lesional skin might identify other similarities besides mTOR signaling. Indeed, targeting mTOR with rapamycin reveres some of the skin changes in Sema4a KO mice, but not skin thickness, so other pathways impacted by Sema4a may be better targets if they could be identified. Utilizing WT→KO chimeras in addition to global KO mice in the experiments in Figures 6-8 would more strongly implicate the separate role of Sema4a in skin vs immune cell populations and might more closely mimic non-lesional psoriasis skin.

We sincerely appreciate your summary and for pointing out the strengths and weaknesses of our study. Although we were unfortunately unable to perform all these experiments due to limitations in our resources, we fully agree with the importance of studying tissue-specific Sema4A KO mice. As an alternative, we compared the IL-17A-producing potential of skin T cells between WT→KO mice and KO→KO mice following 4 consecutive days of IMQ treatment using flow cytometry. The results were comparable between the two groups. Additionally, we performed RNA-seq on the epidermis of WT and Sema4A KO mice. While we did not find similarities between Sema4A KO skin and non-lesional psoriasis except for *S100a8* expression, we will further try to seek for the mechanisms how Sema4A KO skin mimics non-lesional psoriasis skin as a future project.

Although targeting mTOR with rapamycin did not reverse the epidermal thickness in Sema4A KO mice, rapamycin was effective in reducing epidermal thickness in a murine psoriasis model induced by IMQ in Sema4A KO mice. These results suggest potential clinical relevance for treating active, lesional psoriatic skin changes, which would be of interest to clinicians. Thank you once again for your valuable insights.

**Reviewer #2 (Public Review):**
Summary:Kume et al. found for the first time that Semaphorin 4A (Sema4A) was downregulated in both mRNA and protein levels in L and NL keratinocytes of psoriasis patients compared to control keratinocytes. In peripheral blood, they found that Sema4A is not only expressed in keratinocytes but is also upregulated in hematopoietic cells such as lymphocytes and monocytes in the blood of psoriasis patients. They investigated how the down-regulation of Sema4A expression in psoriatic epidermal cells affects the immunological inflammation of psoriasis by using a psoriasis mice model in which Sema4A KO mice were treated with IMQ. Kume et al. hypothesized that down-regulation of Sema4A expression in keratinocytes might be responsible for the augmentation of psoriasis inflammation. Using bone marrow chimeric mice, Kume et al. showed that KO of Sema4A in non-hematopoietic cells was responsible for the enhanced inflammation in psoriasis. The expression of CCL20, TNF, IL-17, and mTOR was upregulated in the Sema4AKO epidermis compared to the WT epidermis, and the infiltration of IL-17-producing T cells was also enhanced.Strengths:Decreased Sema4A expression may be involved in psoriasis exacerbation through epidermal proliferation and enhanced infiltration of Th17 cells, which helps understand psoriasis immunopathogenesis.Weaknesses:The mechanism by which decreased Sema4A expression may exacerbate psoriasis is unclear as yet.

We greatly appreciate your summary and thoughtful feedback on the strengths and weaknesses of our study. In response, we have included the results of additional experiments on IL-23-mediated psoriasis-like dermatitis, which showed that epidermal thickness was significantly greater in KO mice compared to WT mice. When we analyzed the T cells infiltrating the ears using flow cytometry, the proportion of IL-17A producing Vγ2 and DNγδ T cells within the CD3 fraction of the epidermis was significantly higher in Sema4A KO mice, consistent with the results from IMQ-induced psoriasis-like dermatitis. Furthermore, we examined STAT3 expression in the epidermis of WT and Sema4A KO mice using Western blot analysis, and the results were comparable between the two groups. However, the mechanism by which decreased Sema4A expression may exacerbate psoriasis remains unclear. We have added some explanations and presumptions to the limitations section. Thank you once again for your valuable insights.

**Recommendations For The Authors:**

**Reviewer #1 (Recommendations For The Authors):**
Figure 1CWhat statistics were used? The supplemental notes adjusted the P value, what correction for multiple comparisons was utilized? Could the authors instead show logFC for the DEGs between Ctl and L in each cluster? This might be best demonstrated with a volcano plot, highlighting SEMA4A, and other genes known to be DE in psoriasis.

We apologize for not including the detailed analysis methods in the original manuscript submission. We analyzed the scRNA-seq data using Cellxgene VIP with Welch’s t-test. Multiple comparisons were performed using the Benjamini-Hochberg procedure, setting the false discovery rate (FDR) at 0.05. These details are now explained in the MATERIALS AND METHODS section of the resubmitted manuscript. We also added a log2FC-log10 *p*-value graph for the DEGs in keratinocytes between Ctl and L to Figure 1-figure supplement 1D. The log2FC values in keratinocytes, dendritic cells, and macrophages were -0.07, 0.00, and -0.05, respectively. Although the log2FC is low in keratinocytes, the adjusted p-value (*padj*) for Sema4A is 2.83×10-39, indicating a statistically significant difference.

Page 8 Line 111 in the resubmitted manuscript:

“The adjusted p-value (*padj*) for *SEMA4A* in keratinocytes between Ctl and L was 2.83×10-39, indicating a statistically significant difference despite not being visually prominent in the volcano plot, which shows comprehensive differential gene expression in keratinocytes (Figure 1C; Figure 1-figure supplement 1D).”

Page 54: In the Figure legend of Figure 1-figure supplement 1D in the resubmitted manuscript:

“(D) The volcano plot displays changes in gene expression in psoriatic L compared to Ctl.”

Page 30 Line 481 in the resubmitted manuscript: In the “Data processing of single-cell RNA-sequencing and bulk RNA-sequencing” section.

“The data was integrated into an h5ad file, which can be visualized in Cellxgene VIP (K. Li et al., 2022). We then performed differential analysis between two groups of cells to identify differential expressed genes using Welch’s t-test. Multiple comparisons were controlled using the Benjamini-Hochberg procedure, with the false discovery rate set at 0.05 and significance defined as *padj* < 0.05.”

*Figure 2B*

*The results narrative notes WT->WT is comparable to KO->WT. No statistics are given for this comparison. It appears the difference is less than the other comparisons, but still may be significant. Also, in the supplemental for Figure 2B, there appear to be missing columns for the 4 BM chimera groups (columns for WT and KO, but not 4 columns for each donor: recipient pair).*

We sincerely apologize for any confusion. We presented the results of the chimeric mice in Figure 3, and Figure 3-source data 1 shows the 4 BM chimera groups. In Figure 3B, the *p*-value for the comparison between WT->WT mice and KO->WT mice was 0.7988, as indicated in Figure 3-source data 1.

Figure 3BWhile ear skin is not easily obtainable at day 0 for comparison, why not also include back skin at Wk 8? If the back skin epidermis is thicker like the ear skin, it supports the ear skin conclusion and adds a more consistent comparison. If the back skin epidermis is not thicker, what would be the author's explanation as to the why only ear skin epidermis is thicker in KO mice at 8 weeks?

We appreciate and completely agree with the reviewer’s insightful comment. We have added images and dot plots of the back skin at Week 8 in Figure 4B. Since the back skin epidermis is thicker, similar to the ear skin, these results support the conclusion drawn from the ear skin data. Regarding Figure 4C, which shows the expression of Sema4a in the epidermis and dermis of 8-week-old WT mouse ear, we have modified the sentence in the manuscript to ‘the epidermis of WT ear at Week 8’ for clarification.

Page 12 Line 180 in the resubmitted manuscript:

“While epidermal thickness of back skin was comparable at birth (Figure 4B), on week 8, epidermis of Sema4AKO back and ear skin was notably thicker than that of WT mice (Figure 4B), suggesting that acanthosis in Sema4AKO mice is accentuated post-birth.”

Page 47: In the Figure legend of Figure 4B in the resubmitted manuscript:

“(B) Left: representative Hematoxylin and eosin staining of Day 0 back and Wk 8 back and ear. Scale bar = 50 μm. Right: Epi and Derm thickness in Day 0 back (*n* = 5) and Wk 8 back (*n* = 5) and ear (*n* = 8).”

Figures 3C&D, Figures 4 D-FThe figures might be easier to read if some of the data is moved to supplemental, especially in Figure 4, which has 36 panels just in D-F. Conversely, the dLN data is important in establishing the skin microenvironment as important in the accumulation of γδ cells and IL-17 production in the setting of Sema4a KO, so this might be more impactful if moved to the main figure.

We appreciate and agree with your comments. As recommended, we have moved data from Figure 3C and 4D-F to the supplemental section. The dLN data have been moved to the main figure as Figure 4E. This has improved the readability of the figures.

Figure 5 and Figure 6 might work better if combined. The differences in keratinocytes in psoriasis are well-known, so the novelty is how Sema4a KO skin appears to share similar differences. This would be easier to see if compared side-by-side in the same figure. Also, there is an opportunity to show this more rigorously by performing RNA-seq on WT vs Sema4a KO skin. Showing a larger set of DEGs that trend similarly between Ctl/NL psoriasis and WT/Sema4a KO skin in a heatmap would bolster the conclusion that Sema4a deficiency contributes to a psoriasis-like skin defect.

We appreciate your valuable suggestion. Following your recommendation, we have combined Figures 5 and 6 to facilitate a side-by-side comparison. This highlights the similarities between Sema4AKO skin and psoriasis, making it easier to observe differences in keratinocytes. Additionally, we performed RNA-seq on WT and Sema4a KO epidermis (*n* = 3 per group). We analyzed the raw count data using iDEP 2.0 (Ge S.X., *BMC Bioinformatics*, 2018), setting the minimal counts per million to 0.5 in at least one library. Differential gene expression analysis was conducted using DEseq2, with an FDR cutoff of 0.1 and a minimum fold change of 2. As a result, we identified 46 upregulated and 70 downregulated genes in Sema4AKO mice compared to WT mice (see the volcano plot and heat map). However, except for *S100a8*, we did not observe significant expression changes in non-lesional psoriasis-related genes between WT and Sema4AKO mice. In the future, we aim to identify subtle stimuli that could cause gene expression changes between these groups and we would like to perform additional RNA-seq experiments.

**Author response image 2. sa3fig2:** 

Page 48: The Figure title of Figure 5 in the resubmitted manuscript:

“Figure 5: Sema4AKO skin shares the features of human psoriatic NL.”

SEMA4A is not significantly DE between Ctl and NL in the psoriasis RNA-seq data. If a lower expression of SEMA4A in psoriasis skin is a driving part of the phenotype, why is this not observed in the RNA-seq data? Presumably, this could be explained by infiltration of immune cells with increased SEMA4A expression, like in the scRNA-seq data in Figure 1. If so, might it be useful to analyze WT->KO chimera mice similarly to global KO mice in Figures 6-8? This might more accurately reflect what is happening in psoriasis, if epidermal SEMA4A expression is low, but immune expression is not. The KO data on their own nicely show a skin phenotype, but these additional experiments might more closely mimic psoriatic disease and increase the rigor and impact of the study.

We really appreciate your insightful comments. Due to the limitations of the animal experimentation facility, we regret that we are unable to create additional chimeric mice. Although our analysis is limited, we compared IL-17A production from T cells of WT→KO mice and KO→KO mice following 4 consecutive days of IMQ treatment using flow cytometry (see Author response image 3 below; *n* = 6 for WT→KO, *n* = 4 for KO→KO). This comparison revealed that IL-17A production from T cells was comparable, regardless of whether they were derived from WT or Sema4AKO mice, when the skin constituent cells were derived from Sema4AKO. We appreciate the value of your advice, and agree that investigating keratinocyte differentiation and mTOR signaling in the epidermis, using either WT→KO chimeric mice or keratinocyte-specific Sema4A-deficient mice, is a crucial next step in our research.

**Author response image 3. sa3fig3:** 

Figure 8Rapamycin was able to partially reverse the psoriasis-like skin phenotype in Sema4a KO mice. Would rapamycin also be effective in the more severe disease induced by IMQ in Sema4a KO mice? While partially reducing the effect of Sema4a KO on steady-state skin with rapamycin strengthens the link to mTOR dysregulation, it did not change skin thickness. It's unclear if this would be useful clinically for patients with well-controlled psoriasis (NL skin). Would it be useful to reverse active, lesional psoriatic skin changes? Testing this might yield results more relevant to clinicians and patients.

We are grateful for your valuable feedback. Rapamycin showed effectiveness in reducing epidermal thickness in a murine psoriasis model induced by IMQ in Sema4AKO mice. Rapamycin treatment downregulated the expression of *Krt10*, *Krt14*, and *Krt16*. We included these results to Figure 7-figure supplement 2. These results suggest potential clinical relevance for treating active, lesional psoriatic skin changes and may be of interest to clinicians and patients.

Page 17 Line 269 in the resubmitted manuscript:

“Next, we investigated whether intraperitoneal rapamycin treatment effectively downregulates inflammation in the IMQ-induced murine model of psoriasis in Sema4AKO mice (Figure 7-figure supplement 2A). Rapamycin significantly reduced epidermal thickness compared to vehicle treatment (Figure 7-figure supplement 2B). Additionally, rapamycin treatment downregulated the expression of *Krt10*, *Krt14*, and *Krt16* (Figure 7-figure supplement 2C). While the upregulation of *Il17a* in the Sema4AKO epidermis in IMQ model was not clearly modified by rapamycin (Figure 7-figure supplement 2C), immunofluorescence revealed a decrease in the number of CD3 T cells in Sema4AKO epidermis by rapamycin (Figure 7-figure supplement 2D). In the naive states, mTORC1 primarily regulates keratinocyte proliferation, whereas mTORC2 mainly involved in the keratinocyte differentiation through Sema4A-related signaling pathways. Conversely, in the psoriatic dermatitis state, rapamycin downregulated both keratinocyte differentiation and proliferation markers. The observed similarities in *Il17a* expression following treatment with rapamycin and JR-AB2-011, regardless of additional IMQ treatment, suggest that *Il17a* production is not significantly dependent on Sema4A-related mTOR signaling.”

Page 29 Line 461 in the resubmitted manuscript: In the “Inhibition of mTOR” section.

“To analyze the preventive effectiveness of rapamycin in an IMQ-induced murine model of psoriatic dermatitis, Sema4AKO mice were administered either vehicle or rapamycin intraperitoneally from Day 0 to Day 17, and IMQ was topically applied to both ears for 4 days starting on Day 14. Then, on Day 18, ears were collected for further analysis.”

Page 71: Figure 7-figure supplement 2 in the resubmitted manuscript:

“Figure 7-figure supplement 2: Rapamycin treatment reduced the epidermal swelling observed in IMQ-treated Sema4AKO mice.

(A) Experimental scheme. (B) The Epi thickness on Day 18. (*n* = 10 for Ctl, *n* = 12 for Rapamycin). (C) Relative expression of keratinocyte differentiation markers and Il17a in Sema4AKO Epi (*n* = 10 for Ctl, *n* = 12 for Rapamycin). (D) The number of T cells in the Epi (left) and Derm (right), under Ctl or rapamycin and IMQ treatments (*n* = 10 for Ctl, *n* = 12 for Rapamycin). Each dot represents the sum of numbers from 10 unit areas across 3 specimens. **A**-**C**: **p* < 0.05, ***p* < 0.01. NS, not significant.”

**Reviewer #2 (Recommendations For The Authors):**
(1) To know whether the decrease of Sema4A in the epidermis of psoriasis patients is a result or a cause of psoriasis, it is necessary to show how the expression of Sema4A in epidermal cells is regulated. Shouldn't the degree of change in the expression of essential molecules (which is the cause of psoriasis) be more pronounced in L than in NL?

We surveyed transcription factors of human Sema4A using GeneCards and found that NF-κB is the transcription factor most frequently associated with psoriasis. Wang et al. (*Arthritis Res Ther.* 2015) indicated NF-κB-dependent modulation of Sema4A expression in synovial fibroblasts of rheumatoid arthritis. However, since NF-κB expression is reportedly upregulated in psoriasis lesions, other transcription factors may function as key modulators of Sema4A expression in the epidermis.

Although the molecules causing psoriasis remain to be elucidated, we investigated the correlation between the expression of psoriasis-related essential molecules in keratinocytes—such as *S100A7A, S100A7*, *S100A8*, *S100A9*, and *S100A12*—and *SEMA4A* expression in L and NL samples using qRT-PCR. We could not identify a correlation between these molecules and *SEMA4A* expression. We added a note to the limitations section to acknowledge that we were not able to reveal how Sema4A expression is regulated and that we could not determine the relationships between Sema4A expression and the essential molecules upregulated in psoriatic keratinocytes.

Page 21 Line 328 in the resubmitted manuscript:

“We were not able to reveal how Sema4A expression is regulated. Although we showed that downregulation of Sema4A is related to the abnormal cytokeratin expression observed in psoriasis, we could not determine the relationships between Sema4A expression and the essential molecules upregulated in psoriatic keratinocytes.”

(2) Using bone marrow chimeric mice, it has already been reported that hematopoietic cells contain keratinocyte stem cells. Therefore, their interpretation is not supported by the results of their bone marrow chimeric mice experiment, and it is essential to generate keratinocyte-specific Sema4A knockout mice and perform similar experiments to support their interpretation.

We value the reviewer’s insightful comment. We have assessed the expression of *Sema4a* in the epidermis of WT→KO chimeric mice using qRT-PCR. Our findings indicate that *Sema4a* expression levels in the epidermis of these mice are minimal (cycle threshold values of *Sema4a* ranged from 31.9 to not detected in WT→KO chimeric mice, whereas they ranged from 24.5 to 26.2 in WT→ WT mice). Consequently, we believe that the impact of keratinocyte stem cells derived from WT-hematopoietic cells is limited in this model. We appreciate this opportunity to clarify our results and will consider the generation of keratinocyte-specific Sema4A knockout mice for future experiments to further substantiate our interpretation.

Page 11 Line 159 in the resubmitted manuscript:

“Since it has already been reported that bone marrow cells contain keratinocyte stem cells (Harris et al., 2004; Wu, Zhao, & Tredget, 2010), we confirmed that epidermis of mice deficient in non-hematopoietic Sema4A (WT→KO) showed no obvious detection of *Sema4a*, thereby ruling out the impact of donor-derived keratinocyte stem cells infiltrating the host epidermis (Figure 3-figure supplement 1A).”

Page 60: In the Figure legend of Figure 3-figure supplement 1A in the resubmitted manuscript:

“(A) *Sema4a* expression in the Epi of WT→ WT mice and WT→ KO mice (*n* = 8 for WT→ WT, *n* = 7 for WT→ KO).”

(3) Since Sema4A KO mice already have immunological and epidermal cell characteristics similar to psoriasis, albeit weak, it is possible that the nonspecific stimulus of simply topical IMQ may have appeared to exacerbate psoriasis. It is advisable to confirm whether a more psoriasis-specific stimulus, IL-23 administration, would produce similar results.

Thank you for your suggestion. Following your advice, we have analyzed IL-23-mediated psoriasis-like dermatitis. To induce the model, 20 μl of phosphate-buffered saline containing 500 ng of recombinant mouse IL-23 was injected intradermally into both ears for 4 consecutive days. Unlike with the application of IMQ, there was no significant difference in ear thickness. However, H&E staining revealed that the epidermal thickness was significantly greater in KO mice compared to WT mice. Although a longer period of IL-23 induction might result in more pronounced ear swelling, we conducted this experiment over the same duration as the IMQ application experiment to maintain consistency. When we analyzed the T cells infiltrating the ears using flow cytometry, the proportion of IL-17A producing Vγ2 and DNγδ T cells in CD3 fraction in the epidermis was significantly higher in Sema4A KO mice, consistent with the results from IMQ-induced psoriasis-like dermatitis.

The lack of significant difference in ear thickness changes with IL-23 administration might be due to IL-23 administration not reflecting upstream events of IL-23 production.

We consider that in psoriasis, the expression of Sema4A in keratinocytes is likely more important than in T cells. Therefore, it makes sense that the phenotype difference was more pronounced with IMQ, which likely has a greater effect on keratinocytes compared to IL-23.

Page 9 Line 137 in the resubmitted manuscript:

“Though the imiquimod model is well-established and valuable murine psoriatic model (van der Fits et al., 2009), the vehicle of imiquimod cream can activate skin inflammation that is independent of toll-like receptor 7, such as inflammasome activation, keratinocyte death and interleukin-1 production (Walter et al., 2013). This suggests that the imiquimod model involves complex pathway. Therefore, we subsequently induced IL-23-mediated psoriasis-like dermatitis (Figure2-figure supplement 2A), a much simpler murine psoriatic model, because IL-23 is thought to play a central role in psoriasis pathogenesis (Krueger et al., 2007; Lee et al., 2004). Although ear swelling on day 4 was comparable between WT mice and Sema4AKO mice (Figure2-figure supplement 2B), the epidermis, but not the dermis, was significantly thicker in Sema4AKO mice compared to WT mice (Figure2-figure supplement 2C). We found that the proportion of CD4 T cells among T cells was significantly higher in Sema4A KO mice compared to WT mice, while the proportion of Vγ2 and DNγδ T cells among T cells was comparable between them (Figure 2-figure supplement 2D). On the other hand, focusing on IL-17A-producing cells, the proportion of IL-17A-producing Vγ2 and DNγδ T cells in CD3 fraction in the epidermis was significantly higher in Sema4A KO mice, consistent with the results from imiquimod-induced psoriasis-like dermatitis. (Figure 2-figure supplement 2E).”

Page 24 Line 363 in the resubmitted manuscript: In the “Mice” section.

“To induce IL-23-mediated psoriasis-like dermatitis, 20 μl of phosphate-buffered saline containing 500 ng of recombinant mouse IL-23 (BioLegend, San Diego, CA) was injected intradermally into both ears of anesthetized mice using a 29-gauge needle for 4 consecutive days.”

Page 58: In the Figure legend of Figure 2-figure supplement 2 in the resubmitted manuscript:

“IL-23-mediated psoriasis-like dermatitis is augmented in Sema4AKO mice.

(A) An experimental scheme involved intradermally injecting 20 μl of phosphate-buffered saline containing 500 ng of recombinant mouse IL-23 into both ears of WT mice and KO mice for 4 consecutive days. Samples for following analysis were collected on Day 4. (B and C) Ear thickness (B) and Epi and Derm thickness (C) of WT mice and KO mice on Day 4 (*n* = 12 per group). (D and E) The percentages of Vγ3, Vγ2, DNγδ, CD4, and CD8 T cells (D) and those with IL-17A production (E) in CD3 fraction in the Epi (top) and Derm (bottom) of WT and KO ears (*n* = 5 per group). Each dot represents the average of 4 ear specimens. B-E: *p < 0.05, **p < 0.01. NS, not significant.”

(4) How is STAT3 expression in the epidermis crucial in the pathogenesis of psoriasis in Sem4AKO mice?

We appreciate your insightful comment. In our study, given the established role of activated STAT3 in psoriasis, we investigated both total STAT3 and phosphorylated STAT3 (p-STAT3) levels in the naive epidermis of WT and Sema4AKO mice (See the figure below). Our findings indicate that STAT3 activation does not occur in the epidermis of Sema4AKO mice. Therefore, we speculated that the hyperkeratosis observed in Sema4AKO mice is due to aberrant mTOR signaling rather than STAT3 activation. STAT3 may be relevant to other pathways independent of Sema4A signaling, or it may function as a complex with other molecules in the Sema4A signaling.

**Author response image 4. sa3fig4:**